

# Implementing Riverine Biogeochemical Inputs in ECCO-Darwin: a Critical Step Forward for a Pioneering Data-Assimilative Global-Ocean Biogeochemistry Model

Raphaël Savelli[1,2], Dustin Carroll[1,2], Dimitris Menemenlis[2], Jonathan Lauderdale[3], Clément Bertin[2], Stephanie Dutkiewicz[3,4], Manfredi Manizza[5], Anthony Bloom[2], Karel Castro-Morales[6], Charles E. Miller[2], Marc Simard[2], Kevin W. Bowman[2], and Hong Zhang[2]

[1]Moss Landing Marine Laboratories, San José State University, Moss Landing, CA, USA
[2]Jet Propulsion Laboratory, California Institute of Technology, Pasadena, CA, USA
[3]Department of Earth, Atmospheric and Planetary Sciences, Massachusetts Institute of Technology, Cambridge, Massachusetts, USA
[4]Center for Global Change Science, Massachusetts Institute of Technology, Cambridge, Massachusetts, USA
[5]National Institute of Oceanography and Applied Geophysics - OGS, Sgonico, Italy
[6]Deutscher Wetterdienst, Climate and Environment, Offenbach, Germany

**Correspondence:** Raphaël Savelli (raphael.savelli@sjsu.edu)

**Abstract.** Resolving riverine biogeochemical inputs in ocean biogeochemistry models is pivotal for capturing the spatiotemporal variability of nutrients and carbon in coastal regions and in the global ocean. ECCO-Darwin is a pioneering data-assimilative global-ocean biogeochemistry model, which, to date, has focused on the pelagic zone. As a key step towards improving the representation of coastal regions in ECCO-Darwin, we add lateral inputs of carbon, nitrogen, and silica and evaluate the model response with regard to primary production and ocean carbon cycling. We generate riverine inputs by combining point-source freshwater discharge from JRA55-do with the Global NEWS 2 watershed model, accounting for lateral inputs from 5171 watersheds worldwide. While adding carbon and nutrients along with freshwater improves biogeochemical skill in river plume regions and coastal waters, the open-ocean response may be overestimated due to an excess of carbon and nutrients advected offshore. This highlights the need for a more nuanced representation of land-to-ocean and nearshore processes for quantifying how global-ocean primary production and carbon cycling respond to land-to-ocean inputs.

## 1 Introduction

Rivers transport carbon from land to coastal regions as Dissolved Organic Carbon (DOC), Dissolved Inorganic Carbon (DIC), Particulate Organic Carbon (POC), and Particulate Inorganic Carbon (PIC). Nutrients such as phosphorus, nitrogen, and silica, which are essential for phytoplankton growth, are also conveyed by rivers. Terrestrial inorganic carbon and nutrients in streams originate from weathering of the lithosphere and the associated uptake of atmospheric $CO_2$, along with the remineralization of organic matter in streams and/or on land (Suchet and Probst, 1995; Battin et al., 2023). At the same time, most of the refractory part of riverine organic carbon is transported offshore from river mouth regions as it is remineralized at slower turnover rates (Hansell et al., 2004; Holmes et al., 2008; Kaiser et al., 2017).



Once injected into the coastal ocean, riverine carbon (0.7–1 Pg C yr$^{-1}$; Lacroix et al., 2021; Resplandy et al., 2018; Gao et al., 2024; Liu et al., 2024) is buried in coastal sediments, transported into the open ocean and outgassed back to the atmosphere in the form of $CO_2$ (Liu et al., 2024; Regnier et al., 2022; Battin et al., 2023; Gao et al., 2024). Riverine carbon is exchanged with the atmosphere due to the saturation of surface-ocean waters by terrestrial DIC and the remineralization of terrestrial organic matter (Hartmann et al., 2009; Lacroix et al., 2020; Bertin et al., 2023) in shallow, well-mixed water columns. On continental shelves, the outgassing of $CO_2$ driven by terrestrial DIC can also be compensated by the excess of alkalinity relative to DIC (Cai, 2011; Louchard et al., 2021). Concerning nutrients, their injection into the surface ocean can fertilize the growth of photosynthetic organisms in nutrient-limited regions. The subsequent primary production by photosynthetic organisms enhances $CO_2$ uptake by carbon fixation and alkalinity production. Globally, this lateral input increases ocean primary productivity and contributes to an estimated coastal-ocean sink of $\sim 0.25$ Pg C yr$^{-1}$, which is roughly 17% of the global-ocean sink (Cai, 2011; Lacroix et al., 2021; Gao et al., 2023).

While monitoring global riverine inputs to the ocean is challenging due to the necessary tremendous financial/human effort in remote environments, land surface and watershed models can provide spatiotemporally-resolved lateral inputs at global scales (Mayorga et al., 2010; Krinner et al., 2005; Hagemann and Dümenil, 1997; Hagemann and Gates, 2003; Li et al., 2017; Bloom et al., 2020; Gao et al., 2023). Coupled with Global-Ocean Biogeochemical Models (GOBMs), it is thus possible to quantify the response of the coastal- and open-ocean carbon cycle to lateral inputs (Aumont et al., 2001; Lacroix et al., 2021; Mathis et al., 2022; Louchard et al., 2021; da Cunha and Buitenhuis, 2013; Le Fouest et al., 2013; Terhaar et al., 2019; Gao et al., 2023; Bertin et al., 2023; Manizza et al., 2019; Séférian et al., 2020).

By combining (i) a property-conserving ocean circulation reanalysis produced by the Estimating the Circulation and Climate of the Ocean (ECCO) project, (ii) an ocean ecology model constructed by the Massachusetts Institute of Technology (MIT) Darwin Project, (iii) a representation of ocean carbon chemistry, and (iv) property-conserving data assimilation tools developed by ECCO, we have developed a state-of-the-art global-ocean biogeochemistry state estimate that we call ECCO-Darwin. ECCO-Darwin provides data-constrained, global-ocean estimates of ocean circulation, sea ice, marine ecology, and biogeochemistry; and it has demonstrated skill in representing space-time variability in global-to-regional carbon cycling (Carroll et al., 2020, 2022; Bertin et al., 2023). But, to date, ECCO-Darwin studies have focused on the pelagic zone and have not included biogeochemical river discharge, i.e., carbon and nutrient inputs at the global scale (Carroll et al., 2020). In this study, we 1) add point-source lateral inputs of carbon and nutrients to ECCO-Darwin globally and 2) evaluate the model response of air-sea $CO_2$ flux and primary production to riverine biogeochemical inputs during 2000–2019. Implementing biogeochemical river input into ECCO-Darwin is a critical stepping stone in the development of this data-constrained modeling framework. The modeling advances described herein will allow for further integration of the Land-Ocean Aquatic Continuum (LOAC) in ECCO-Darwin; a pivotal pathway for understanding the response of ocean biogeochemistry to terrestrial inputs.



## 2  Methods

### 2.1  The ECCO-Darwin Ocean Biogeochemistry State Estimate

The ECCO-Darwin ocean biogeochemistry state estimate is extensively described in Brix et al. (2015), Manizza et al. (2019), and Carroll et al. (2020, 2022, 2024). For the ECCO-Darwin model presented in this study, ocean physics (circulation, temperature, salinity, and sea ice) are provided by a prerelease of the ECCO Version 4 release 5 (V4r5) global-ocean and sea-ice data synthesis. A detailed overview of ECCO V4 is available in Forget et al. (2015) while specific details pertaining to V4r5 are being made available in ECCO et al. (2024). Horizontal discretization is based on a Lat-Lon-Cap-90 (LLC90) configuration of the MIT general circulation model (MITgcm; Marshall et al., 1997a, b). Nominal horizontal grid spacing is $1°$ but telescopes to $\sim$33 km meridionally near the Equator and to $\sim$55 km in the Arctic Ocean. The vertical discretization consists of 50 z-levels, ranging from 10-m thickness in the top 7 levels to 450 m at maximum depth of 6 km. ECCO V4 uses a third-order, direct-space-time tracer advection scheme in the horizontal and an implicit third-order upwind scheme in the vertical; a time step of 3600 s is used. Vertical mixing is parameterized using the Gaspar–Grégoris–Lefevre (GGL) mixing-layer turbulence closure and convective adjustment scheme (Gaspar et al., 1990). ECCO V4 assimilates physical observations via the adjoint method (Wunsch et al., 2009; Wunsch and Heimbach, 2013). Importantly, ECCO V4 is a property-conserving ocean reanalysis, that is, contrary to reanalyses that are based on sequential estimation methods, ECCO V4 satisfies model equations exactly for the complete period of optimization (1992–2020 for V4r5). This characteristic makes ECCO V4 uniquely well-suited for ocean ecology and biogeochemistry applications.

Daily river discharge in the present configuration is based on the Japanese 55-year atmospheric Reanalysis (JRA55) for driving ocean–sea-ice models (JRA55-do). JRA55-do river discharge is computed based on the Catchment-based Macro-scale Floodplain (CaMa-Flood) global river routing model and on adjusted runoff from the land component of JRA55 (Suzuki et al., 2018; Tsujino et al., 2018; Feng et al., 2021). JRA55-do point source freshwater runoff was added to ECCO V4r5 as a freshwater flux in the surface ocean (first vertical level) at the closest corresponding ECCO V4r5 grid cell along the coastal periphery. The freshwater flux was adjusted according to the difference in grid cell area between JRA55-do ($0.25°$x $0.25°$) and ECCO V4r5. A complete evaluation of ocean physics from ECCO V4r5 compared to observations can be found in the Supporting Information and in Feng et al. (2021).

ECCO V4r5 ocean physics were coupled online with the MIT Darwin Project ecosystem model described in Brix et al. (2015). The ecosystem model solves 39 prognostic variables, including carbon, nitrogen, phosphorus, iron, silica, oxygen, and alkalinity. The model simulates their respective cycle from inorganic pools to living/dead matter of plankton organisms and the subsequent remineralization, all driven by the ocean physics. The carbonate chemistry is solved with the method of Follows et al. (2006). Plankton species consist of five large-to-small functional phytoplankton types (diatoms, other large eukaryotes, *Synechococcus*, and low- and high-light adapted *Prochlorococcus*) and two zooplankton types. Atmospheric $CO_2$ partial pressure at sea level (A$p$CO$_2$) from the National Oceanic and Atmospheric Administration Marine Boundary Layer Reference product (Andrews et al., 2014) was used to drive air-sea $CO_2$ fluxes computed by the model according to Wanninkhof (2014). Atmospheric iron dust is deposited at the ocean surface based on the monthly climatology of Mahowald et al. (2009).





**Table 1.** Sensitivity experiments and associated solutes.

| Experiment Name | Solutes |
|---|---|
| **Baseline** | - |
| **DC$_{run}$** | t$_{DOC}$ + t$_{DIC}$ |
| **DIC$_{run}$** | t$_{DIC}$ |
| **DN$_{run}$** | t$_{DON}$ + t$_{DIN}$ |
| **DIN$_{run}$** | t$_{DIN}$ |
| **DSi$_{run}$** | t$_{DSi}$ |
| **ALL$_{run}$** | t$_{DOC}$ + t$_{DIC}$ + t$_{DON}$ + t$_{DIN}$ + t$_{DSi}$ |

In the water column, particulate organic matter sinks at prescribed velocities and is removed at the sea bottom to limit the
accumulation of particulates on the seafloor. ECCO-Darwin assimilates biogeochemical observations using a Green's Functions
optimization approach (Menemenlis et al., 2005); the optimization methodology and associated data constraints are extensively
described in Carroll et al. (2020). The ECCO-Darwin solution was previously published using an LLC270 (1/3°) ECCO
solution (Zhang et al., 2018) and monthly climatological freshwater runoff forcing from Fekete et al. (2002). Here, we introduce
a new 1°-version of ECCO-Darwin with daily point-source freshwater runoff from January 1992 to December 2019 (hereinafter
our "Baseline" simulation) and also conduct a suite of perturbation experiments (Table 1) where we add various riverine
biogeochemical input components to assess the primary productivity and carbon cycle response. Except for these changes,
our simulations use the same initial conditions, parameter settings, and forcings as in Carroll et al. (2020). To account for
biogeochemical spin-up in the perturbation runs, the following analysis was performed for the last 20 years of simulation, from
January 2000 to December 2019 (Supporting Information Figures S3 and S4).

## 2.2  Model Evaluation

We compared simulated surface-ocean partial pressure in $CO_2$ ($p$CO$_2$) and air-sea $CO_2$ fluxes in Baseline with state-of-the-art
products based on the Surface Ocean $CO_2$ Atlas (SOCAT; Bakker et al., 2016; Sabine et al., 2013). We used the monthly
$p$CO$_2$ and air-sea $CO_2$ fluxes MPI-SOM-FFN v2023 (Landschützer et al., 2016; Jersild et al., 2023) and Copernicus Marine
Environment Monitoring Service (CMEMS; Chau et al., 2022) climatologies computed from neural network-based clustering
algorithms. In addition, we used the monthly atmospheric $CO_2$ inversion Jena Carboscope v2023 (Rödenbeck et al., 2013)
based on high-precision measurements from the Gridded Fossil Emissions Dataset (GridFED; Jones et al., 2021) and SOCAT
(Bakker et al., 2016; Sabine et al., 2013). These products were interpolated on the LLC90 grid from January 2000 to December 2019. Grid cells covered by sea-ice (concentration $> 0\%$) were discarded from the model-data evaluation, based on the
percentage of sea-ice cover simulated by ECCO-Darwin. Furthermore, we compared data-based products (MPI-SOM-FFN,
Copernicus CMEMS, Jena Carboscope, and mean of all products) with co-located output from the Baseline and **ALL$_{run}$** sim-




ulations to quantify the change in surface-ocean $p$CO$_2$ and air-sea $CO_2$ flux associated with the addition of biogeochemical inputs.

## 2.3 Biogeochemical River Discharge Product

In addition to the Baseline simulation, we conducted a suite of sensitivity experiments (Table 1) where we added terrestrial DOC ($t_{DOC}$), DIC ($t_{DIC}$), total alkalinity ($t_{ALK}$), dissolved inorganic nitrogen ($t_{DIN}$), dissolved organic nitrogen ($t_{DON}$), and dissolved silica ($t_{DSi}$), henceforth referred to as riverine inputs in this study. Except for $t_{DIC}$ and $t_{ALK}$, riverine inputs are provided by the Global Nutrient Export from WaterSheds 2 (NEWS 2; Mayorga et al., 2010) model. The method for computing our daily point-source input, which is then used as forcing in ECCO-Darwin along the coastal periphery of the global ocean, is detailed below.

Global NEWS 2 uses statistical and mechanistic relations at the watershed scale to compute annual-mean freshwater discharge and riverine inputs based on natural and anthropogenic sources, with 6292 individual watersheds delineated according to the global river systems dataset from Vörösmarty et al. (2000). Global NEWS 2 $t_{DIN}$ was partitioned into nitrite ($NO_2^-$), nitrate ($NO_3^-$), and ammonium ($NH_4^+$), according to the mean fraction of each species concentration relative to the total DIN concentration from the GLObal RIver CHemistry Database (GLORICH; Hartmann et al., 2014). The $NO_2^-$:DIN, $NO_3^-$:DIN, and $NH_4^+$:DIN ratios were estimated to be 0.02, 0.65, and 0.33, respectively.

$t_{DIC}$ inputs were computed using an empirical relation between freshwater discharge and gross $CO_2$ consumption from rock weathering, as described in Li et al. (2017, equation 9). $CO_2$ consumption by rock weathering over each Global NEWS 2 watershed was estimated based on the freshwater discharge and the basin-dominant lithology (Amiotte Suchet et al., 2003). $t_{ALK}$ inputs were computed using an ALK:DIC ratio (0.98) based on the mean total ALK compared to DIC from GLORICH. The remineralization rate for terrestrial and marine DOC equals 1 over 100 days. We used Global NEWS 2 outputs for the year 2000 as representative of present-day carbon and nutrient inputs (Mayorga et al., 2010). Riverine inputs were compared against observations from literature and the Arctic Great Rivers Observatory (ArcticGRO) water-quality monitoring network in the Arctic region (Holmes et al., 2012; Tank et al., 2023) (Supporting Information Figure S2 and Table S2).

Global NEWS 2 river mouth locations were associated with JRA55-do grid points exhibiting the closest annual-mean freshwater discharge in 2000 within an euclidean distance of 5°. The top 100 largest rivers (by watershed extent) from Global NEWS 2 were imposed on JRA55-do grid points as a function of distance only. In total, 5171 river mouths were associated with JRA55-do grid points. For each discharge point, riverine input concentrations (g m$^{-3}$) from the associated river were estimated by dividing the load by the annual volume of freshwater from Global NEWS 2; the concentration was then multiplied by the corresponding daily-mean freshwater flux from JRA55-do (m s$^{-1}$) to obtain a daily flux (g m$^{-2}$ s$^{-1}$). Riverine inputs were adjusted according to the grid-cell-area difference between JRA55-do and ECCO V4r5. Then, these biogeochemical inputs were added as point-source discharge along with riverine freshwater flux (Table 2 and Supporting Information Figure S1). Due to extreme values in our automated Global NEWS 2-derived computation for the Amazon River, the DIC input for this system was set to a more realistic, literature-mean of 2.54 Tmol yr$^{-1}$ (da Cunha and Buitenhuis, 2013; Probst et al., 1994; Li et al., 2017) (for more details, see Appendix A). Riverine $t_{DOC}$, $t_{DIN}$, $t_{DON}$, and $t_{DSi}$ inputs agree well with existing estimates in





Table 2. t$_{DIC}$ lateral input from rivers was estimated according to Amiotte Suchet et al. (2003); Mayorga et al. (2010); Li et al. (2017), resulting in a t$_{DIC}$ input of 381.81 Tg C yr$^{-1}$ to the ocean, which is in general agreement with recent studies in Table 2. More data-model comparisons are available in Supporting Information Table S2 and Figure S2.

### 2.4 Sensitivity Experiments and Analysis

Sensitivity experiments consisted of adding riverine inputs separately or together, along with freshwater runoff (Table 1). t$_{ALK}$ was always added along with t$_{DIC}$ in relevant experiments. We analyzed monthly-mean model fields both in the coastal ocean (limits set by the furthest point from the coastline, either the 1000-m isobath or a distance of 300 km; 58 x 10$^6$ km$^2$) and open ocean (300 x 10$^6$ km$^2$) from 2000–2019. We also evaluated the sensitivity of ocean carbon cycling in three specific regions that receive large volumes of freshwater and biogeochemical inputs from major river systems (Lacroix et al., 2020): the Arctic

Ocean (ARCT, 22 x 10$^6$ km$^2$), Tropical Atlantic (TROP-ATL, 77 x 10$^6$ km$^2$), and Southeast Asia (SE-ASIA, 62 x 10$^6$ km$^2$). Coastal and open-ocean boundaries are delineated by the black line shown in Figure 1a.

### 3 Results

### 3.1 Model-product Evaluation

Overall, ECCO-Darwin Baseline surface-ocean $p$CO$_2$ compares reasonably well with the Jena Carboscope, MPI-SOM-FFN,

and Copernicus CMEMS data-based products (Figure 1). The largest differences are concentrated along the coastal periphery and near large river mouths (i.e., Amazon, Paraná, Congo, Ganges, Yangtze, Amur), where ECCO-Darwin Baseline underestimates surface-ocean $p$CO$_2$ (Figure 1i). Additionally, the data-based products exhibited lower surface-ocean $p$CO$_2$ compared to ECCO-Darwin Baseline (Figure 1i) in the Arctic Ocean and near the periphery of Antarctica; regions where observations are highly limited in space and time.

Figure 2 shows a comparison of time-mean ECCO-Darwin air-sea CO$_2$ flux (-2.58 Pg C yr$^{-1}$), Jena Carboscope v2023 (-2.11 Pg C yr$^{-1}$), MPI-SOM-FFN v2023 (-2.04 Pg C yr$^{-1}$), and Copernicus CMEMS (-1.97 Pg C yr$^{-1}$) products during 2000–2019. Compared to the product mean, ECCO-Darwin air-sea CO$_2$ flux yields a stronger ocean CO$_2$ uptake (+0.5 Pg C yr$^{-1}$, Figure 2i). Overall, the spatial distribution of source/sink patterns in the global ocean was well captured by ECCO-Darwin Baseline. However, the products displayed stronger CO$_2$ outgassing in the North Pacific and Atlantic Oceans, the

Arabian Sea, and the Southern Ocean (Figure 2i).

The addition of riverine biogeochemical runoff in ALL$_{run}$ increased the surface-ocean $p$CO$_2$ and air-sea CO$_2$ flux deviation from data-based products in the Arctic Ocean, Bay of Bengal, Sea of Japan, and Caribbean Sea (Figure 3a,d). However, the deviation decreased locally near large river mouths (Amazon, Orinoco, Paraná, Yangtze, Ganges rivers) (Figure 3a,d). While the input of dissolved carbon was responsible for the decrease of the deviation in river plumes, it results in a large increase

in surface-ocean $p$CO$_2$ deviation in the Arctic Ocean, especially on the Siberian Shelf (Figure 3b). Dissolved nitrogen was primarily responsible for an increase in deviation from the data-based products, especially in the vicinity of large rivers (Figure



**Table 2.** Riverine inputs.

| Domain | Input (Tg yr$^{-1}$) | ALL$_{run}$ | Literature Value |
|--------|----------------------|-------------|------------------|
| **Global** | $t_{DOC}$ | 170.1 | 130–300 [1] |
| | $t_{DIC}$ | 381.8 | 320–520 [1] |
| | $t_{DON}$ | 11.7 | 11.8 [1] |
| | $t_{DIN}$ | 23.3 | 17–22.8 [1] |
| | $t_{DSi}$ | 139.7 | 158–171 [1] |
| **ARCT** | $t_{DOC}$ | 22.6 | 34–37.7 [2] |
| | $t_{DIC}$ | 56.8 | 57 [2] |
| | $t_{DON}$ | 1.4 | 0.05–0.84 [2] |
| | $t_{DIN}$ | 1.1 | 0.04–0.43 [2] |
| | $t_{DSi}$ | 12.6 | 11.4 [2] |
| **TROP-ATL** | $t_{DOC}$ | 67.2 | 46 [3] |
| | $t_{DIC}$ | 78.1 | 50 [3] |
| | $t_{DON}$ | 4.2 | N/A |
| | $t_{DIN}$ | 4.5 | 15.3 [3] |
| | $t_{DSi}$ | 44.9 | 53 [3] |
| **SE-ASIA** | $t_{DOC}$ | 36.6 | N/A |
| | $t_{DIC}$ | 163.8 | 40 [4] |
| | $t_{DON}$ | 2.6 | N/A |
| | $t_{DIN}$ | 10.6 | 2.1–8.4 [4*] |
| | $t_{DSi}$ | 41.5 | N/A |

[1] (Drake et al., 2018; Mayorga et al., 2010; Frings et al., 2016; Sharples et al., 2017; Lacroix et al., 2020; Tivig et al., 2021; Tian et al., 2023; Li et al., 2017; Lacroix et al., 2020; Battin et al., 2023; Tian et al., 2023; Liu et al., 2024)

[2] (Manizza et al., 2011; Tank et al., 2012; Holmes et al., 2012; Le Fouest et al., 2013)

[3] (Cotrim da Cunha et al., 2007; Araujo et al., 2014)

[4] (Singh and Ramesh, 2011; Li et al., 2014; Wang et al., 2020; Nishina et al., 2021; Piao et al., 2012; Patra et al., 2013)

[*] Computed from sum of regional estimates.



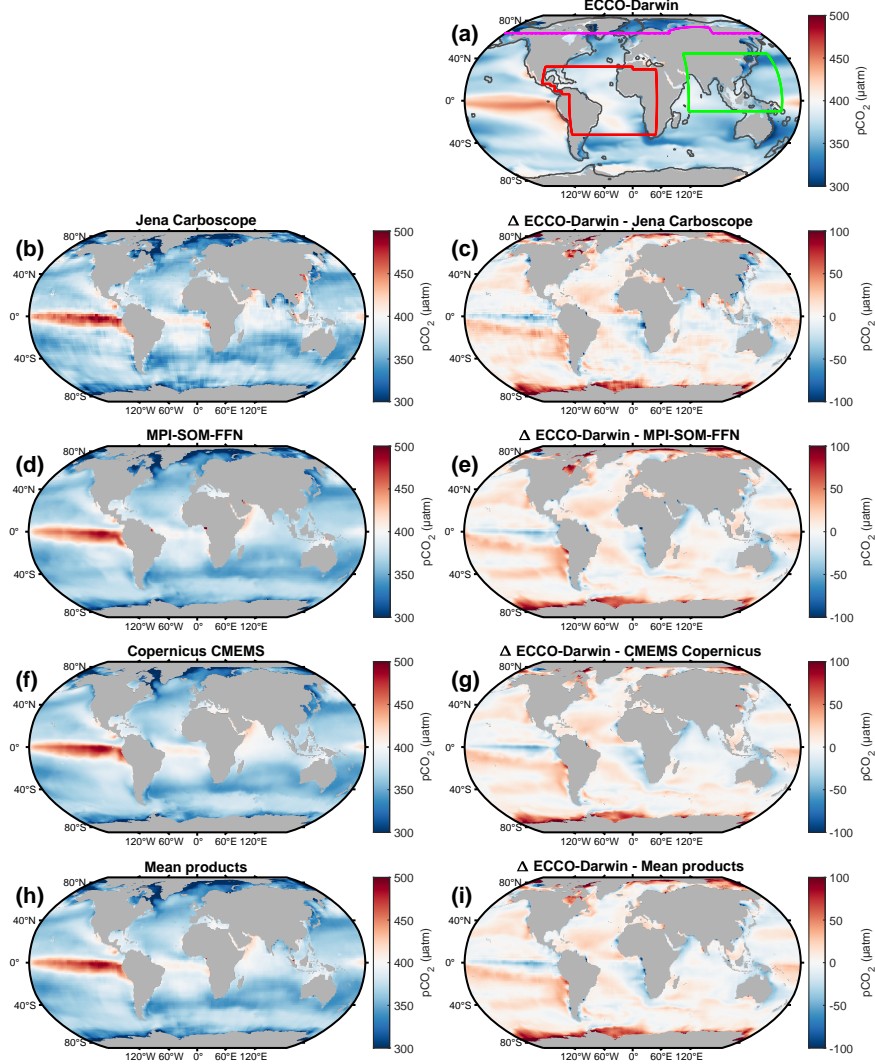

**Figure 1.** Climatological global-ocean surface-ocean $p$CO$_2$ for (a) ECCO-Darwin Baseline, (b) Jena Carboscope, (d) MPI-SOM-FFN, (f) Copernicus CMEMS, and (h) mean of all data products. Panels (c), (e), (g), and (i) correspond to the difference between ECCO-Darwin Baseline and each data product. All fields shown are time means from January 2000 to December 2019. In (a), colored boundaries correspond to domains used for regional analysis of the Arctic Ocean (ARCT, violet line), Tropical Atlantic (TROP-ATL, red line), and Southeast Asia (SE-ASIA, green line). The black line delineates the coastal ocean from the open ocean, which is set by the furthest point from the coastline of either a 300-km distance or the 1000-m isobath. MPI-SOM-FFN, Jena Carboscope, and CMEMS Copernicus products were interpolated on the LLC90 grid.



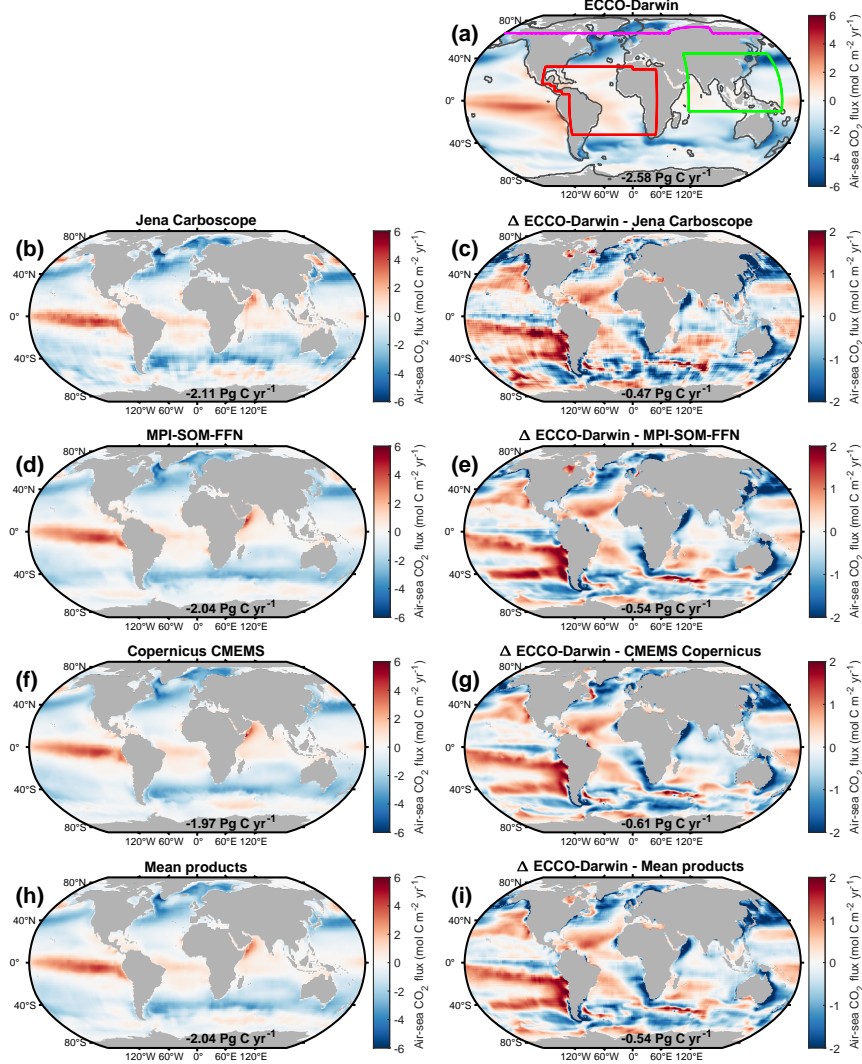

**Figure 2.** Climatological global-ocean air-sea $CO_2$ flux for (a) ECCO-Darwin Baseline, (b) Jena Carboscope, (d) MPI-SOM-FFN, (f) Copernicus CMEMS, and (h) the mean of all data-based products. Panels (c), (e), (g), and (i) correspond to the difference between ECCO-Darwin Baseline and each product. Positive values represent $CO_2$ outgassing (red colors); negative values represent uptake (blue colors). All fields shown are time means from January 2000 to December 2019. In (a), colored boundary lines correspond to domains used for regional analysis of the Arctic Ocean (ARCT, violet line), Tropical Atlantic (TROP-ATL, red line), and Southeast Asia (SE-ASIA, green line). The black line delineates the coastal ocean from the open ocean, which is set by the furthest point from the coastline of either a 300-km distance or the 1000-m isobath. MPI-SOM-FFN, Jena Carboscope, and CMEMS Copernicus products were interpolated on the LLC90 grid.



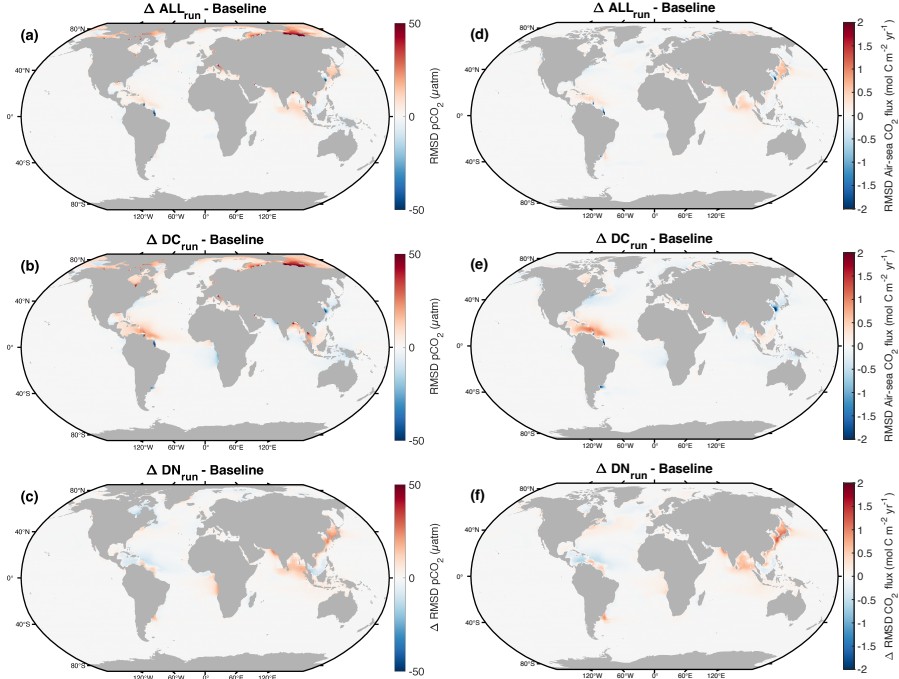

**Figure 3.** Change in root-mean-square deviation (RMSD) of $p$CO$_2$ (left panels) and air-sea CO$_2$ flux (right panels) between ECCO-Darwin Baseline and (a,d) ALL$_{run}$, (b,e) DC$_{run}$, and (c,f) DN$_{run}$ compared to the data-based product mean (MPI-SOM-FFN, Jena Carboscope, and CMEMS Copernicus). Positive values represent an increase of the deviation (red colors); negative values show a smaller deviation (blue colors). All fields shown are time means from January 2000 to December 2019.

3c,f); this increase was particularly marked in the Bay of Bengal and Sea of Japan. In the Caribbean Sea, the moderate increase in deviation in ALL$_{run}$ resulted from the compensatory effort of dissolved carbon (outgassing) and nitrogen (uptake from enhanced biological productivity) (Figure 3).





## 3.2 Climatological Global Analysis

**Table 3.** Air-sea $CO_2$ flux and Net Primary Production (NPP) for each experiment in the coastal, open, and global ocean. Positive values represent $CO_2$ outgassing; negative values are uptake.

| Domain | Experiment | $CO_2$ Flux $(Pg\ C\ yr^{-1})$ | NPP $(Pg\ C\ yr^{-1})$ |
|---|---|---|---|
| **Coastal Ocean** | Baseline | −0.68 | 3.8 |
| | $ALL_{run}$ − Baseline | +0.04 | +0.3 |
| | $DC_{run}$ − Baseline | +0.1 | 0.0 |
| | $DN_{run}$ − Baseline | −0.07 | +0.3 |
| | $DSi_{run}$ − Baseline | −0.01 | +0.01 |
| **Open Ocean** | Baseline | −1.90 | 20.9 |
| | $ALL_{run}$ − Baseline | −0.01 | +0.3 |
| | $DC_{run}$ − Baseline | +0.1 | 0.0 |
| | $DN_{run}$ − Baseline | −0.1 | +0.3 |
| | $DSi_{run}$ − Baseline | −0.01 | +0.01 |
| **Global Ocean** | Baseline | -2.58 | 24.7 |
| | $ALL_{run}$ − Baseline | +0.03 | +0.6 |
| | $DC_{run}$ − Baseline | +0.22 | 0.0 |
| | $DN_{run}$ − Baseline | −0.17 | +0.6 |
| | $DSi_{run}$ − Baseline | −0.01 | +0.01 |

The addition of dissolved carbon, nitrogen, and silica in $ALL_{run}$ led to a small increase in $CO_2$ outgassing of 0.03 Pg C yr$^{-1}$ compared to Baseline, globally (Table 3 and Figure 4a). The majority of $CO_2$ outgassing driven by riverine inputs (0.04 Pg C yr$^{-1}$) occurs in the coastal ocean (Table 3 and Figures 4a and 5a). In the open ocean, riverine inputs slightly increased $CO_2$ uptake by 0.01 Pg C yr$^{-1}$ (Table 3 and Figures 4a and 5a).

In $ALL_{run}$, the small net change in air-sea $CO_2$ flux results from compensation between the effects of riverine carbon and nitrogen, as $DC_{run}$ and $DN_{run}$ experiments result in elevated $CO_2$ outgassing and uptake, respectively (Table 3 and Figure 5a). In $DC_{run}$, the increase in ocean carbon, and hence $pCO_2$ due to riverine inputs reduces the ocean's capacity to take up atmospheric $CO_2$, resulting in a net $CO_2$ outgassing of 0.22 Pg C yr$^{-1}$ (Table 3 and Figure 5a). In $DN_{run}$, the increase of nutrients in the





euphotic zone elevates phytoplankton productivity. The additional uptake of carbon by phytoplankton decreased surface-ocean

DIC, resulting in an additional global-ocean $CO_2$ uptake of 0.17 Pg C $yr^{-1}$ (Table 3 and Figure 5a).

$CO_2$ outgassing driven by riverine carbon ($DC_{run}$) was dominated by $t_{DOC}$ (70%), with a smaller contribution from $t_{DIC}$ (30%) (Figure 5a). While outgassing driven by carbon inputs were compensated by uptake due to nitrogen in the open ocean, $CO_2$ uptake due to nitrogen was 36% lower than carbon-input-driven coastal outgassing, resulting in a global-ocean $CO_2$ uptake that was reduced by 0.03 Pg C $yr^{-1}$ (i.e., increased outgassing) in $ALL_{run}$ compared to Baseline (Figure 5a).

Dissolved carbon, nitrogen, and silica inputs in $ALL_{run}$ result in a Net Primary Production (NPP) increase of 0.6 Pg C $yr^{-1}$ (+2%) compared to Baseline (Table 3 and Figure 4b). $t_{DIN}$ ($DIN_{run}$) input contributed to 70% and 86% of the $CO_2$ uptake and NPP increase simulated in $DN_{run}$, respectively (Figure 5b). The total increase of NPP in $ALL_{run}$ from riverine inputs was equally distributed between the coastal and open ocean (0.3 Pg C $yr^{-1}$ for each) (Table 3 and Figure 5b). However, the increase of NPP was larger in the coastal ocean (+5.2 g C $m^{-2}$ $yr^{-1}$, +7%) compared to the open ocean (+1 g C $m^{-2}$ $yr^{-1}$,

+1%), relative to their surface area.

## 3.3 Climatological Regional Analysis

In Baseline, ARCT results in a $CO_2$ uptake of roughly 0.21 Pg C $yr^{-1}$. When carbon, nitrogen, and silica inputs are added in $ALL_{run}$, ARCT $CO_2$ uptake reduces by 0.02 Pg C $yr^{-1}$, with the majority of the response (80%) in the coastal ocean (Table 4, Figures 4a and 5a). Input-driven $CO_2$ outgassing was dominated by the input of $t_{DOC}$ in ARCT (Figure 5a). In Baseline,

ARCT NPP was 0.22 Pg C $yr^{-1}$, with a similar magnitude in the coastal and open ocean. Adding nitrogen input into ARCT increased coastal NPP by 40% (Figure 5b).

Carbon, nitrogen, and silica inputs resulted in a TROP-ATL $CO_2$ outgassing of 0.02 Pg C $yr^{-1}$ compared to Baseline (0.10 Pg C $yr^{-1}$). This imbalance results from $CO_2$ outgassing driven by dissolved carbon, which was 30% larger than the uptake due to increased phytoplankton productivity from dissolved nitrogen (Figure 5b). 85% of $CO_2$ outgassing in $DC_{run}$ is driven by

$t_{DOC}$ (Figure 5a). $t_{DOC}$ indirectly drives $CO_2$ outgassing by being remineralized to DIC, which increases DIC concentration and $pCO_2$ in the ocean and reduces the ocean's capacity to take up atmospheric $CO_2$. Most of the $CO_2$ outgassing driven by $t_{DOC}$ occurs in the open ocean (Figures 4a and 5a). In Baseline, NPP in TROP-ATL was 3.18 Pg C $yr^{-1}$. The increase in NPP driven by riverine $t_{DIN}$ occurs predominantly in the open ocean ($\sim$60%) compared to the coastal ($\sim$40%) zone (Figures 4b and 5b).

SE-ASIA has a $CO_2$ uptake of 0.30 Pg C $yr^{-1}$ in Baseline, while $CO_2$ uptake increases by 0.02 Pg C $yr^{-1}$ in $ALL_{run}$ (Table 4 and Figure 4a). Carbon and nitrogen inputs drive compensating air-sea $CO_2$ fluxes in the coastal ocean (respective contribution of $\pm 0.03$ Pg C $yr^{-1}$). However, in the open ocean, the nitrogen input-driven increase in NPP and associated $CO_2$ uptake is two times higher than carbon input-driven outgassing — leading to an overall imbalance and resulting in net $CO_2$ uptake in SE-ASIA (Figure 5a). $t_{DIN}$ is responsible for 85% of biological $CO_2$ uptake (Figure 5a). NPP in SE-ASIA without riverine

inputs is 3.3 Pg C $yr^{-1}$. In $ALL_{run}$, NPP increases by 0.30 Pg C $yr^{-1}$ due to elevated $t_{DIN}$ in both the open and coastal ocean (Figure 5b).





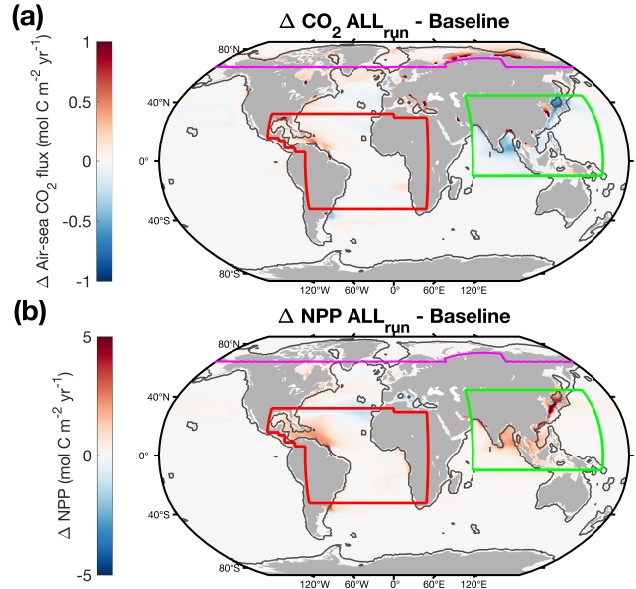

**Figure 4.** Global-ocean (a) air-sea $CO_2$ flux and (b) NPP driven by riverine inputs in $ALL_{run}$. In panel (a), positive values represent more $CO_2$ outgassing (red colors), and negative values represent more uptake (blue colors) compared to Baseline. Fields represent time-mean values from January 2000 to December 2019. Colored lines on maps show domains used for regional analysis. The black line delineates the coastal ocean from the open ocean.

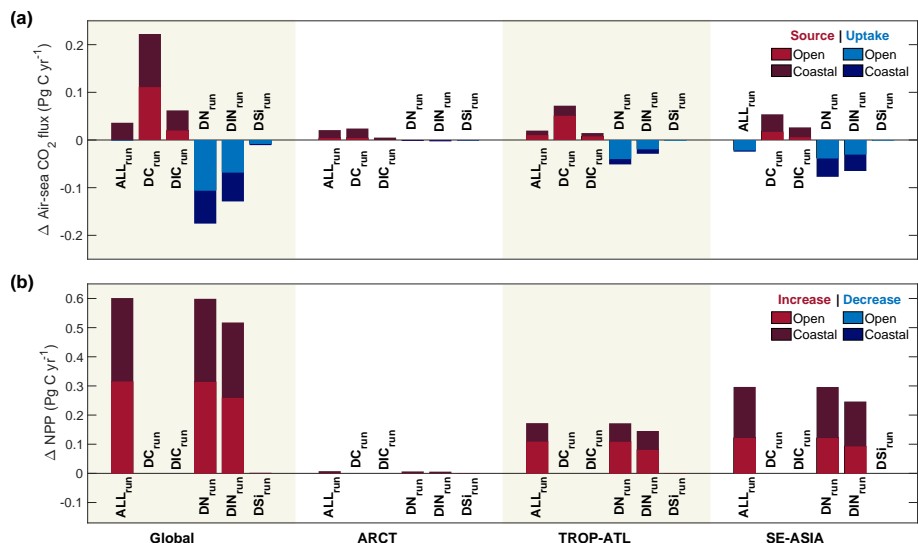

**Figure 5.** Domain-integrated differences in (a) air-sea $CO_2$ flux and (b) NPP driven by exports in each sensitivity experiment. Differences were computed from time-mean fields from January 2000 to December 2019.



## 4  Discussion

### 4.1  Riverine Inputs in ECCO-Darwin

Compared to state-of-the-art observation-based products, our current ECCO-Darwin set-up based on ECCO V4r5 exhibits
similar results to the versions described in Carroll et al. (2020, 2022). Baseline depicts a time-mean global-ocean $CO_2$ sink of
2.58 Pg C yr$^{-1}$ during 2000–2019. This is in relatively good agreement with MPI-SOM-FFN v2023 (-2.04 Pg C yr$^{-1}$; Land-
schützer et al., 2016; Jersild et al., 2023), Jena Carboscope v2023 (-2.11 Pg C yr$^{-1}$; Rödenbeck et al., 2013), and Copernicus
CMEMS (-1.97 Pg C yr$^{-1}$; Chau et al., 2022) products over the same period. The largest differences occur along the coastal
periphery, especially near large rivers mouths where Baseline, with its lack of riverine inputs and freshwater discharge only, un-
derestimates surface-ocean $p$CO$_2$ compared to the data-based products. Lower Baseline surface-ocean $p$CO$_2$ and air-sea $CO_2$
fluxes in these regions are driven by freshwater input only. In the absence of associated biogeochemistry, freshwater discharge
dilutes chemical species in the coastal ocean, decreasing the salinity, the concentration of DIC, and the alkalinity in surface
waters. This highlights the need to include coupled freshwater and biogeochemical discharge in GOBMs, as associated carbon
and nutrients can compensate for the freshwater-only dilution effect. In the marginal ice zone of high latitudes, the data-based
products depict lower surface-ocean $p$CO$_2$ and air-sea $CO_2$ fluxes compared to Baseline. As the data-products are primarily
computed from statistical/mechanistic models based on the SOCAT database, the sparse observational coverage can be a source
of error and uncertainty in these regions. We note that in regions such as the Antarctic Continental Shelf and the Arctic Ocean,
which have extensive seasonal sea-ice cover, the SOCAT database coverage is limited (Bakker et al., 2016; Sabine et al., 2013).

In ALL$_{run}$, the deviation of ECCO-Darwin surface ocean $p$CO$_2$ and air-sea $CO_2$ fluxes from the data-based products is re-
duced near large river plumes (Amazon, Paraná, Congo, Ganges, Yangtze, Amur) by the addition of dissolved carbon input. In
these regions, $CO_2$ outgassing driven by carbon inputs compensates for the dilution effect from freshwater discharge. However,
in the Bay of Bengal, Sea of Japan, Canadian Arctic Archipelago, Caribbean Sea, and Siberian Shelf, model skill decreases
with riverine inputs. While the data-based products converge on a $CO_2$ sink over the Arctic basin, adding dissolved carbon in
ALL$_{run}$ and the associated $CO_2$ outgassing increases the model mismatch in this region. However, large gaps in SOCAT data
coverage in the Arctic Ocean may be responsible for this discrepancy, especially on the Siberian Shelf. In the Bay of Bengal
and Sea of Japan, the ocean $CO_2$ sink is increased by the higher uptake from enhanced-NPP in response to the addition of
nitrogen input in ALL$_{run}$. In the Caribbean Sea, the slight increase in mismatch is associated with carbon input from Tropical
Atlantic rivers, such as the Amazon and Orinoco. Overall, adding carbon inputs along with freshwater runoff substantially
improves ECCO-Darwin's representation of surface-ocean $p$CO$_2$ and air-sea $CO_2$ flux in river plume regions and compensates
for the dilution effect. In regions with moderate carbon inputs, model performance compared to the data-based products de-
creased with the addition of nitrogen. In regions with large carbon and/or nitrogen inputs (TROP-ATL/ARCT and SE-ASIA,
respectively), the model deviates from data-based products in terms of surface-ocean $p$CO$_2$ and air-sea $CO_2$ flux in the open
ocean. In these regions, biogeochemical inputs might be overestimated due to the absence of LOAC processes in the model (i.e.,
parameterizations of estuarine mixing and biogeochemical processes) and of a more realistic representation of organic matter
remineralization, allowing for the advection of excess dissolved carbon and nutrients into the open ocean. In ARCT, the gaps





in observational data make it difficult to assess if the addition of pan-Arctic inputs increases model skill; however, a regional set-up of ECCO-Darwin in the Beaufort Sea has recently demonstrated increased model skill when including biogeochemical inputs (Bertin et al., 2023).

Baseline captures similar spatial patterns of NPP compared to the model ensemble of the REgional Carbon Cycle Assess-
ment and Processes Phase-2 (RECCAP-2) project that aims at constraining present-day ocean carbon from observation-based estimates, inverse models, and GOBMs (Doney et al., 2024). Many uncertainties remain regarding global-ocean NPP estimates from remote sensing (due to uncertainty in algorithms) and models (due to different conceptual model architecture). Overall, NPP in Baseline (24.7 Pg C yr$^{-1}$) lies in the lower bound of the wide range depicted by the RECCAP-2 model ensemble (25–57 Pg C yr$^{-1}$; Doney et al., 2024) and remote-sensing algorithms (43–68 Pg C yr$^{-1}$; Behrenfeld and Falkowski, 1997;
Silsbe et al., 2016; Carr et al., 2006; Marra et al., 2003; Behrenfeld et al., 2005). This relatively low NPP results primarily from strong iron limitation in the High-Nutrient, Low-Chlorophyll (HNLC) regions in ECCO-Darwin (Carroll et al., 2020). The strong surface-ocean stratification and the weaker winter convection limit the replenishment of nutrients in the euphotic zone. Nevertheless, global-ocean NPP estimates will improve from enhanced space-time coverage of NPP measurements and associated key variables such as chlorophyll, light, nutrients, optical properties, and cell physiology (Bendtsen et al., 2023). An
integration of environmental variables along with NPP measurements will greatly reduce models' spread and mismatch with synoptic in-situ observations. The implementation of a radiative transfer package (Dutkiewicz et al., 2019) in the next version of ECCO-Darwin, for which development is already underway, will permit the assimilation of direct ocean-color observations (remotely-sensed reflectance) and improve the model's estimate of global-ocean NPP.

We acknowledge that adding lateral inputs of freshwater, carbon, and nutrients in ECCO-Darwin Baseline can result in an
additional source of spin-up and drift in the model simulations. As Baseline and sensitivity experiments are based on the same physical solution, the drift associated with the addition of freshwater is removed from our analysis; however, biogeochemical inputs may be an additional source of drift in the simulations presented in this study (Supporting Information Figures S3 and S4). As the model sensitivity to inputs might require thousands of years to reach an equilibrium, the simulated ocean response to riverine inputs might be limited by the ECCO-Darwin multi-decadal timescale (Lacroix et al., 2020). While the use of a
Green's Functions-based optimization has been shown to reduce spin-up and drift in previous ECCO-Darwin solutions (Brix et al., 2015; Carroll et al., 2020), it will be necessary to optimize a new ECCO-Darwin solution that includes biogeochemical runoff to select the initial conditions and model parameters that will minimize model-data misfit (i.e., cost) and reduce spin-up drift — a focus of ongoing work. We note that the next version of ECCO-Darwin aims to include optimization controls of inputs ratio (DIC:ALK, $NO_2^-$:DIN, $NO_3^-$:DIN, and $NH_4^+$:DIN), allowing us to optimize riverine inputs based on remotely-sensed and
in-situ ocean observations.

## 4.2 Impact of Dissolved Carbon and Nutrient Inputs in ECCO-Darwin

In this study, carbon input drives a $CO_2$ outgassing of 0.22 Pg C yr$^{-1}$, while nitrogen input drives a $CO_2$ uptake of 0.17 Pg C yr$^{-1}$ from enhanced primary productivity, which primarily occurs in the coastal ocean. Combined, carbon and nutrients inputs in ALL$_{run}$ are limited to a 0.03 Pg C yr$^{-1}$ $CO_2$ outgassing; lower than literature estimates (Table 4). Our multi-decadal




**Table 4.** Change in air-sea $CO_2$ flux and NPP driven by riverine inputs. Positive values represent an increase in $CO_2$ outgassing or primary production; negative values represent an increase in $CO_2$ uptake or a decrease in primary production.

| Domain | $\Delta CO_2$/NPP | $ALL_{run}$ (Tg C yr$^{-1}$) | Literature Value (Tg C yr$^{-1}$) |
|---|---|---|---|
| **Global** | $\Delta CO_2$ | +30 | +110 [1] |
| | $\Delta$ NPP | +600 | +600–3900 [1] |
| **ARCT** | $\Delta CO_2$ | +22.8 | +0.6–20 [2*] |
| | $\Delta$ NPP | +7.4 | +58 [2] |
| **TROP-ATL** | $\Delta CO_2$ | +20 | -5–20 [3**] |
| | $\Delta$ NPP | +170 | +80–400 [3] |
| **SE-ASIA** | $\Delta CO_2$ | -24.7 | N/A |
| | $\Delta$ NPP | +296 | +100 [4] |

[1] (Tivig et al., 2021; Cotrim da Cunha et al., 2007)

[2] (Manizza et al., 2011; Terhaar et al., 2021)

[3] (da Cunha and Buitenhuis, 2013; Louchard et al., 2021)

[4] (Tivig et al., 2021)

[*] Effect of t$_{DOC}$ only

[**] Lower bound is for smaller domain in western TROP-ATL

estimates do not include realistic representations of estuarine, blue carbon, and bottom-sediment processes. Therefore, our results do not include air-sea $CO_2$ fluxes associated with these land-to-ocean components.

Adding nutrient inputs, such as inorganic nitrogen and silica, increases global-ocean marine NPP by 0.6 Pg C yr$^{-1}$ compared to Baseline. The addition of t$_{DIN}$ and t$_{DON}$ also increased ocean NPP by 0.6 Pg C yr$^{-1}$ in the model described by Tivig et al. (2021) (Table 4). The modeling study of Lacroix et al. (2020) also depicted a 3% increase in ocean NPP due to the addition of nutrient inputs. In our study, the increase of NPP driven by riverine inputs was stronger in the coastal ocean compared to the open ocean, relative to their respective surface areas. This is consistent with the recent study of Mathis et al. (2024), which demonstrates the role of increased nutrient inputs in driving stronger biological carbon fixation and, thus, an enhanced $CO_2$ sink in the coastal ocean during the last century.

As $ALL_{run}$ deviates more from the data-based products in terms of surface-ocean $pCO_2$ and air-sea $CO_2$ flux in TROP-ATL, ARCT, and SE-ASIA compared to Baseline, the response of the ocean carbon cycle (source and uptake) to riverine inputs might be overestimated due to the advection of excess elements to the open ocean. This may be due to our fixed DOC remineralization rate (100 days), which does not account for terrestrial-originating components with a faster degradation rate (labile to semi-labile), and the absence of a LOAC parameterization to account for estuarine and near-shore processes. For example, the deviation from data-based products on the Siberian Shelf in $ALL_{run}$ may be driven by an excess of carbon





reaching the ocean. Across the Arctic LOAC, permafrost DOC may be degraded and outgassed back to the atmosphere further upstream (river, estuary, river plume) compared to ECCO-Darwin, while our riverine inputs are directly injected into the coastal ocean (Spencer et al., 2015; Bertin et al., 2025). In TROP-ATL, DOC from the Amazon river is expected to be more stable in the coastal ocean (up to hundreds of years; Louchard et al., 2021) and is not released back to the atmosphere in the vicinity of the Caribbean Sea, as depicted by the deviation of ALL$_{run}$ from data-based products in this region. In SE-ASIA, excess inputs

of nitrogen reaching the Bay of Bengal or Sea of Japan may drive model-data mismatch in this region, as the model lacks a LOAC parameterization and especially representation of estuaries where nitrogen can be consumed upstream by biological activity (Cai, 2011).

     Assuming that carbon and/or nutrient inputs from each watershed are routed completely and instantaneously to the ocean is a source of model error, as losses and gains occur through the LOAC, especially in estuaries. Sharples et al. (2017) estimated

that 25% of global DIN input was removed on continental shelves through biological uptake, denitrification, and anaerobic oxidation. Current GOBMs and Earth System Models (ESMs) used in IPCC Assessment Reports compute the amount of carbon introduced to coastal grid cells (i.e, the lateral input) from reference watersheds or land-surface models that do not resolve the transport and transformation of carbon through the LOAC and, especially, estuaries and associated blue carbon pools (i.e., salt marshes and mangroves; Mayorga et al., 2010; Ciais et al., 2014; Lacroix et al., 2020; Ward et al., 2020). While

coastal wetlands, estuaries, and continental shelves are a pivotal filter of carbon and biogeochemical elements, their action on reactive species has yet to be included in most GOBMs (Cai, 2011).

     In Baseline, ARCT uptakes 213.9 Tg C yr$^{-1}$ of atmospheric $CO_2$; this may be an overestimate of the ARCT $CO_2$ sink, as recent estimates from modeling, atmospheric inversions, and $p$CO$_2$-based products range from 91–116 Tg C yr$^{-1}$ (Yasunaka et al., 2023) (Supporting Information Figure S5); although observations are highly limited in this region. Riverine inputs reduce

this $CO_2$ uptake by 20 Tg C yr$^{-1}$ and result in ECCO-Darwin having a more consistent Arctic Ocean's $CO_2$ sink compared to previous estimates (Manizza et al., 2011; Mortenson et al., 2020; Yasunaka et al., 2023) (Supporting Information Figure S6). In ARCT, carbon inputs dominate the response of air-sea $CO_2$ fluxes, with riverine t$_{DOC}$ responsible for 80% of $CO_2$ outgassing (+10%). In Terhaar et al. (2019), $CO_2$ outgassing increases by 90% when riverine t$_{DOC}$ was doubled. However, Terhaar et al. (2019) used an instantaneous remineralization rate for DOC, resulting in rapid outgassing in the coastal region compared to

our results.

     In addition, nutrient inputs also contribute to the Arctic Ocean's carbon sink as they fertilize coastal waters. NPP in the Arctic Ocean increased by 3% (+7.4 Tg C yr$^{-1}$) in ALL$_{run}$ compared to Baseline. In Terhaar et al. (2019), the doubling of riverine nutrients (+2.3 Tg N yr$^{-1}$) leads to an 11%-increase of NPP, where our study suggests a 3%-increase for an equivalent riverine input. Recent estimates by Terhaar et al. (2021) suggest that riverine nutrients support up to 15% (+58 Tg C yr$^{-1}$)

of marine NPP in the ARCT, in agreement with estimates by (Le Fouest et al., 2013, 2015) (Table 4). Therefore, biological $CO_2$ uptake driven by riverine nitrogen and its capacity to compensate $CO_2$ outgassing in ARCT might be underestimated in our study. The functional phytoplankton types in our global model are not representative of the Arctic Ocean, and the lack of nitrogen and silica -limited taxa might hinder the model response to riverine nitrogen and silica (Ardyna and Arrigo, 2020).





In Baseline, TROP-ATL is a source of $CO_2$ to the atmosphere (0.10 Pg C yr$^{-1}$), which agrees with both data-based products
(Landschützer et al., 2016; Jersild et al., 2023; Rödenbeck, 2005) (0.04–0.08 Pg C yr$^{-1}$) and GOBM results (da Cunha and
Buitenhuis, 2013; Louchard et al., 2021) (0.03–0.04 Pg C yr$^{-1}$). We note that previous studies show an input-driven increase
in $CO_2$ uptake of 0.005 and 0.02 Pg C yr$^{-1}$ when adding biogeochemical runoff in TROP-ATL and western TROP-ATL,
respectively (da Cunha and Buitenhuis, 2013; Louchard et al., 2021) (Table 4). However, in our simulations, the addition of
riverine inputs in ALL$_{run}$ enhanced the source of $CO_2$ to the atmosphere (+0.02 Pg C yr$^{-1}$) (Table 4). Contrary to the estimates
of Louchard et al. (2021), which include physical effects associated with freshwater, such as enhanced upper-ocean stratification
and gas solubility, our baseline simulation already includes these processes. Therefore, our set of experiments cannot isolate
and quantify the impact of freshwater discharge on ocean biogeochemistry. In addition to nitrogen, Louchard et al. (2021)
also included inorganic phosphorus exports and a regionally-adjusted plankton ecosystem, e.g., by including a nitrogen-fixing
phytoplankton functional type, which increased the model's capability to resolve the biological pump and hence $CO_2$ uptake.

SE-ASIA is a sink of atmospheric $CO_2$ in Baseline (0.3 Pg C yr$^{-1}$). Combining air-sea $CO_2$ budgets for the different regions
composed of SE-ASIA values from literature (East-Pacific, Indonesian seas, and North Indian Ocean (without including Oman
and Somalian upwelling regions), we estimate an ocean carbon uptake of ∼0.2 Pg C yr$^{-1}$ for the entire SE-ASIA domain
(Kartadikaria et al., 2015; De Verneil et al., 2021; Zhong et al., 2022; Hood et al., 2023). The net air-sea $CO_2$ exchange
balance driven by riverine inputs in SE-ASIA results in a carbon uptake of 0.02 Pg C yr$^{-1}$ in ALL$_{run}$. Compared to ARCT
and TROP-ATL, carbon uptake in SE-ASIA is enhanced by a large increase in marine NPP (+0.30 Pg C yr$^{-1}$, +9%) driven by
t$_{DIN}$ input. In Tivig et al. (2021), the simulated increase of NPP in response to riverine nitrogen was roughly 0.1 Pg C yr$^{-1}$
in Asia, with the strongest increase in the Yellow Sea, similar to our results (Table 4). Locally, adding riverine biogeochemical
runoff also drives a source of $CO_2$ to the atmosphere, which is primarily limited to near river mouth locations in SE-ASIA.
In the Yellow Sea and the Northern Bay of Bengal, close to the Yangtze and Ganges Rivers, the addition of riverine inputs at
preindustrial levels in an ocean model also drove a $CO_2$ outgassing in Lacroix et al. (2020). Noticeably, in our simulations, the
addition of carbon inputs switches the northern Bay of Bengal from a carbon uptake to a source, as suggested by Hood et al.
(2023). Similarly, the addition of t$_{DOC}$ in an ocean model of the Sunda Shelf Sea drives a $CO_2$ outgassing by 3.1 Tg C yr$^{-1}$
from 2013–2022 (Mayer et al., 2025). Most importantly, as nitrogen inputs play a critical role in the SE-ASIA ocean carbon
response, they need to be better constrained by a more extensive suite of observational data (Supporting Information Figure
S2).

### 4.3 Recommendations for a More Realistic Representation of River-driven Carbon Cycling in ECCO-Darwin

Our study is a critical step forward in the development of the data-assimilative ECCO-Darwin framework for estimating how
the ocean's physical-biogeochemical-ecological state responds to present-day terrestrial biogeochemical inputs. In this section,
we elaborate on the necessary and consequential simplifications made in this study. In light of these simplifications, we also
describe some next-step ECCO-Darwin model improvements that are planned or currently underway.

We limited our analysis to t$_{DOC}$, t$_{DIC}$, t$_{ALK}$, t$_{DIN}$, t$_{DON}$, and t$_{DSi}$. However, rivers are a significant source of phosphorus
and iron inputs that are pivotal for ocean biogeochemistry (Krachler et al., 2005; Tagliabue et al., 2017; Duhamel et al., 2021;




Savenko and Savenko, 2021). Furthermore, the stoichiometric ratio between nitrogen and phosphorus is critical for shaping ocean ecology (Redfield et al., 1963; Beusen and Bouwman, 2022). Phosphorus inputs can increase NPP in phosphorus-limited

coastal waters, if not limited by other nutrients (Giraud et al., 2008), which could result in the consumption of additional nutrients in the coastal ocean and could therefore limit the excess of nutrients advected to the open ocean as depicted by our study. Additionally, nitrogen fixation by diazotrophs and, thus, the efficiency of the biological pump is reduced by the absence of a dynamic nitrogen-to-phosphorus ratio from the coincident addition of phosphorus and nitrogen inputs in regions where the limitation by phosphorus prevails, such as in the Tropical Atlantic (Wang et al., 2019; Browning and Moore, 2023).

Consequently, our results may misestimate the regional-scale and global-ocean $CO_2$ sink/source 1) due to the absence of phosphorus or iron inputs in the sensitivity experiments, as they might be limiting for ocean NPP relative to the excess of nitrogen inputs, and 2) the excess of nutrients advected to the open ocean.

Additionally, $t_{ALK}$ inputs were based on a global-mean, constant ALK:DIC ratio (0.98). We note that the GLORICH database used to compute this ALK:DIC ratio has relatively good coverage over the American continent; however, Eurasia

and Africa remain underrepresented (Hartmann et al., 2014). As such, the ALK:DIC ratio can vary substantially over regional and time scales. The lack of this spatially-granular information in our simulated riverine inputs may misrepresent $t_{ALK}$ inputs and the ALK-driven buffering capacity of simulated river plumes (Dubois et al., 2010; Tank et al., 2012; Mol et al., 2018; Ghosh et al., 2021; Gomez et al., 2023; Terhaar et al., 2019). While in estuaries the absence of ALK relative to DIC leads to higher $pCO_2$ in upper-ocean waters and enhanced $CO_2$ outgassing in the coastal zone, rivers also result in an excess of ALK

relative to DIC on continental shelves, which can reduce ocean $pCO_2$ through buffering and, thus, facilitate $CO_2$ uptake (Cai et al., 2010; Louchard et al., 2021).

In the present study, riverine particulate matter 1) rapidly sinks to the seafloor near river mouths, and 2) once at the seafloor, sinking particulates in the model are removed (at a rate equivalent to the sinking rate) to limit the unrealistic accumulation of particulates at depth. Remineralization of sinking particulates associated with riverine inputs and enhanced marine biomass

could be an additional source of dissolved nutrients and carbon to the upper ocean through vertical mixing or upwelling mechanisms; ultimately affecting the air-sea $CO_2$ exchange depicted by the model in the coastal zone. In our current set-up, particulates from riverine-boosted production may be removed at the sediment-water interface too quickly, considering that most of the impact from riverine inputs occurs along the coast in shallow waters. Development to add a diagenetic sediment model in ECCO-Darwin is currently underway (RADI) to provide a more holistic representation of the global-ocean carbon

sink. (Sulpis et al., 2022).

In our model set-up, the global-ocean net $CO_2$ outgassing driven by riverine inputs reflects the stronger effect of riverine carbon on the solubility pump ($CO_2$ source) compared to the change in the $CO_2$ uptake associated with the increase in NPP. This imbalance can result from overestimating $CO_2$ outgassing driven by carbon inputs, especially $t_{DOC}$. First, assuming that watershed-wide carbon and/or nutrient inputs are fully routed to the ocean is a misrepresentation, as losses and gains occur

through the LOAC (Cai, 2011). Second, $t_{DOC}$ is degraded in coastal waters at different rates depending on its origin and subsequent labile fraction (Holmes et al., 2008; Wickland et al., 2012; Shen et al., 2012; Lønborg et al., 2020). In the present study, in addition to not accounting for refractory and labile fractions of $t_{DOC}$, marine and terrestrial DOC are remineralized at





the same rate (100 days). Overall, this could lead to an overestimation of t$_{DOC}$ remineralization and thus ocean $CO_2$ outgassing
due to the subsequent excess of DIC; a limitation that also exists in other GOBMs due to undifferentiated remineralization

rates. While recent modeling studies include separate pools of refractory and labile t$_{DOC}$ with different remineralization rates
at regional scales (Louchard et al., 2021; Gibson et al., 2022; Bertin et al., 2023), the nature of t$_{DOC}$ needs to be better
accounted for in GOBMs (such as in Aumont et al. (2001)). For instance, the Amazon River — the largest global source of
riverine t$_{DOC}$ to the ocean — contributes to almost 50% (+0.014 Pg C yr$^{-1}$) of the global-ocean $CO_2$ outgassing in our study.
However, t$_{DOC}$ from the Amazon River shows strong stability in the coastal ocean and is transported from the continental

margin to the open ocean (Medeiros et al., 2015; Louchard et al., 2021). Increasing the refractory pool of Amazon t$_{DOC}$ could,
therefore, decrease $CO_2$ outgassing in our simulations. Nonetheless, the time scale of t$_{DOC}$ remineralization remains difficult
to constrain as observation-based estimates contain large variability in reported values (Holmes et al., 2008; Wickland et al.,
2012; Shen et al., 2012).

## 5 Conclusion and Perspectives

ECCO-Darwin is an innovative, open-source, ocean biogeochemistry state estimate and modeling infrastructure that is gaining
increasingly wider use for ocean ecology and carbon cycle studies. Unlike Global Ocean Biogeochemistry Models (GOBMs),
ECCO-Darwin assimilates both physical and biogeochemical observations and hence is more consistent with data-based re-
constructions. Unlike data-based reconstructions and reanalyses that employ sequential estimation methods, ECCO-Darwin
solutions satisfy model equations exactly and can therefore be used to explore the individual processes that control primary

production and ocean carbon sequestration. To date, however, ECCO-Darwin has lacked a representation of biogeochemi-
cal inputs from land-ocean exchanges. In this manuscript, we present a new version of ECCO-Darwin that includes daily,
point sources of freshwater and associated biogeochemical inputs, accounting for terrestrial carbon and nutrients routed to the
ocean from 5171 watersheds globally. This is a critical step forward towards improved representation of Land-Ocean Aquatic
Continuum (LOAC) processes in ECCO-Darwin.

As a test of this new capability, this study highlights how the ECCO-Darwin ocean state estimate changes when adding
riverine biogeochemical inputs. While these results are a promising step forward in the development of an improved data-
constrained modeling framework for estimating the global carbon cycle, next-step model improvements, such as a better
representation of water-sediment exchanges, more accurate remineralization rates of organic matter, and improved parame-
terizations of LOAC processes, are necessary. The present ECCO-Darwin Baseline and the perturbation experiments described

in this paper, with their relatively coarse horizontal resolution compared to previous versions (Carroll et al., 2022), offer an
advantage in computational cost and storage for global-ocean studies. However, the Baseline's global optimization has limita-
tions in representing some regional-scale processes. In light of the land-to-ocean model improvements described in this study,
future work is needed to optimize the model and better represent riverine inputs, coastal ecosystems, and DOC lability. Nev-
ertheless, our results pave the way for improved coastal ECCO-Darwin simulations, as the global-ocean version is now being

used for more proficient regional cut-outs (Bertin et al., 2023, 2025) and downscaling efforts, which make possible improved



comparisons and assimilation of land-to-ocean in-situ monitoring network data, such as ArcticGRO (Holmes et al., 2012; Tank et al., 2023).

New satellite missions that cover the LOAC and coastal zone will also enable improved ECCO-Darwin model-data fusion. For instance, the complexity of optical properties and their large space-time variability require high-resolution sensors to capture the actual constituents of ocean biogeochemistry. The recently launched Plankton, Aerosol, Cloud, ocean Ecosystem (PACE) satellite mission (https://pace.oceansciences.org/) and its onboard high-resolution spectrometer will bring new insights into the water quality of coastal regions near river mouths. Combined, with river discharge retrievals from the Surface Water and Ocean Topography (SWOT, https://swot.jpl.nasa.gov/), harmonized, multi-platform spaceborne observations of ocean color and terrestrial inputs of freshwater will pave the way for an integrated Earth Observation System and assimilation of these data streams into ECCO-Darwin, ultimately leading to improved understanding of the land-to-ocean connection and its impact on the global carbon budget.

*Code and data availability.* ECCO-Darwin model output is available at the ECCO Data Portal: http://data.nas.nasa.gov/ecco/. Model code and platform-independent instructions for running the ECCO-Darwin simulations used in this paper and generating runoff forcing are available from the ECCO-Darwin GitHub website: https://github.com/MITgcm-contrib/ecco_darwin/blob/master/v05/1deg_runoff and https://github.com/MITgcm-contrib/ecco_darwin/tree/master/code_util/LOAC/GlobalNews, respectively. Compiled outputs and model code (version on 05/25/2025) used in this study are available at: https://doi.org/10.5281/zenodo.15512392 (Savelli, 2025).

## Appendix A: Amazon River Runoff Set-up

As we computed riverine nutrient inputs from the combination of Global NEWS 2 loads with JRA55-DO runoff, Global NEWS 2 river concentrations must be co-located with JRA55-DO grid points exhibiting the closest annual discharge to avoid under or overestimation of nutrient loads. In the case of the Amazon River, where freshwater and nutrient loads are extreme, we manually assigned the river mouth location from Global NEWS 2 to the corresponding JRA55-DO grid point. Additionally, when using equation in Li et al. (2017, equation 9), the DIC load from the Amazon river was overestimated and was therefore set to a mean literature value of 2.54 Tmol yr$^{-1}$ (da Cunha and Buitenhuis, 2013; Probst et al., 1994; Li et al., 2017).

*Author contributions.* Conceptualization: RS, DM, and DC. Data curation: RS. Funding acquisition: DM, MS, DC, and CEM. Software: RS, DM, DC, SD, and HZ. Evaluation: RS, DC, HZ, and CB. Formal analysis: RS, DM, DC, and JL. Investigation: RS, DM, DC, SD, JL, CB, CEM, MM, KCM, and AB. Project administration: RS, DM, MS, DC, KWB, and CEM. Resources: DM, MS, DC, SD, and CEM. Supervision: DM, MS, and DC. Visualization: RS and DC. Writing—review & editing: RS, DM, DC, SD, JL, CB, CEM, KWB, MS, KCM, and AB.





*Competing interests.* The authors declare no competing interests.

465 *Acknowledgements.* This research was conducted at Moss Landing Marine Laboratories (San José State University). A portion of this research was carried out at the Jet Propulsion Laboratory, California Institute of Technology, under a contract with the National Aeronautics and Space Administration (80NM0018D0004). Support from the Carbon Cycle Science (CCS) and Carbon Monitoring Systems (CMS) programs is acknowledged. High-end computing resources were provided by the NASA Advanced Supercomputing (NAS) Division of the Ames Research Center. Government sponsorship acknowledged. © 2024. All rights reserved



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
