# Peer review of "Implementing Riverine Biogeochemical Inputs in ECCO-Darwin: A Sensitivity Analysis of Terrestrial Fluxes in a Data-Assimilative Global Ocean Biogeochemistry Model"

_EGUsphere, 2025_

## Author Comment (AC1)

We thank the referees for their constructive feedback. We agree with all major comments. Our study needed a clearer message and scope, along with a better assessment of its limitations, particularly regarding missing processes, spin-up, and stoichiometry. We now explicitly position this work as a sensitivity analysis using an existing optimized ECCO-Darwin to quantify the response of the ocean carbon cycle to riverine inputs, rather than to deliver a fully optimized solution that includes coastal processes.

To address concerns about nutrient stoichiometry and phytoplankton limitation, we streamlined the experiment set by adding a single dedicated nutrient simulation that includes both phosphorus and iron inputs. This configuration directly responds to referee feedback by accounting for their combined supply relative to global N:P:Si:Fe stoichiometry. Regarding model spin-up and equilibration timescales, we acknowledge that multi-decadal simulations cannot fully equilibrate riverine nutrient perturbations, especially in the Arctic, and we clarify that such equilibration is required for comparison with the long-term river loop. We further highlight that spin-up limitations will be directly addressed in future work, as we are currently developing a new adjoint-optimized ECCO-Darwin configuration (v06) that includes fully-coupled riverine biogeochemical tracers and other coastal and benthic processes.

In summary, we accept the referee's broader point that a more complete and optimized configuration of ECCO-Darwin will be necessary for comparison with long-term cycling of riverine inputs; this manuscript is an important technical milestone towards that effort Finally, while we recognize suggestions to further expand the analysis, we have deliberately chosen to keep the manuscript focused solely on the implementation of riverine inputs and avoid adding additional layers of complexity, as it is already dense. Our intention in submitting to GMD is for this work to serve as a technical foundation that can be leveraged for more comprehensive scientific exploration in future studies using ECCO-Darwin.

**Referee #1**

In their manuscript, Savelli et al. describe their implementation of river fluxes in the ECCO-Darwin model and evaluate the performance of the model after this implementation. The paper is well written, the methods are generally sound, and the analysis of the effects of river fluxes in the model is well reflected and seems robust. The manuscript is also well suited for GMD. While I think the paper is close to publication, it might lack a bit of really novel aspects, since the implementation of river

fluxes has been described in a few GOBMs in recent years, as cited in the study. I believe it would be relatively straightforward to add a few more interesting aspects originating from the implementation into the ECCO-Darwin model, specifically, and would strongly recommend doing this (phytoplankton species shifts? More detailed process-based explanations of divergent FCO2 responses in the different regions?). I also have a few major points that should be clarified and I hope they will also improve the manuscript.

I would not refer to the implementation as a critical one, as in the title, since the effects of the implementation shown for the global ocean are shown to be quite limited here. While it would be a different case if the model was focusing on coastal fluxes, there does not seem to be any other advancements in this direction presented in this paper.

We agree that the global effect of the implementation is limited in this study. However, we consider the implementation critical for the ECCO-Darwin model and future land-to-ocean model development and studies. Until now, ECCO-Darwin lacked a realistic representation of lateral fluxes of carbon and nutrients, and including these sources is an important step forward for the ECCO-Darwin community, but also for the ocean modeling community in general, as more ocean biogeochemistry models should account for terrestrial carbon and nutrients. Even if the contribution of river carbon and nutrients is small at the global scale, it remains significant for coastal and regional budgets and, thus, is pivotal for future studies using ECCO-Darwin. For this reason, we believe it is critical to move forward with this new ECCO-Darwin capability. We changed the title to make it more reflective of both referees' comments:

Implementing Riverine Biogeochemical Inputs in ECCO-Darwin: A Sensitivity Analysis of Terrestrial Fluxes in a Data-Assimilative Global Ocean Biogeochemistry Model

I think the introduction could be improved. While the important points are there in my opinion, it reads a bit disconnected. I would try to streamline and especially underline the following points: 1. Rivers transport carbon, nutrients and alkalinity to the ocean, 2. These transports are central for biogeochemistry and biological life in coastal regions. They also can affect open ocean biogeochistry due to offshore transport. 4. Models of the current generation do not well represent these transports and their implications.

We modified the introduction to make it more streamlined:

"Rivers transport carbon from land to coastal regions as Dissolved Organic Carbon (DOC), Dissolved Inorganic Carbon (DIC), Particulate Organic Carbon (POC), and Particulate Inorganic Carbon (PIC), along with nutrients such as phosphorus, nitrogen, and silica, which are essential for phytoplankton growth. Terrestrial inorganic carbon

and nutrients in streams originate from weathering of the lithosphere and the associated uptake of atmospheric CO2, along with the remineralization of organic matter in streams and/or on land (Suchet and Probst, 1995; Battin et al., 2023).

Riverine carbon (0.7–1 Pg C yr-1; Lacroix et al., 2021; Resplandy et al., 2018; Gao et al., 2024; Liu et al., 2024) can be buried in coastal sediments, transported into the open ocean, and outgassed back to the atmosphere in the form of CO2 (Liu et al., 2024; Regnier et al., 2022; Battin et al., 2023; Gao et al., 2024). This carbon is transferred to the atmosphere due to the saturation of surface-ocean waters by terrestrial DIC and the remineralization of terrestrial organic matter (Hartmann et al., 2009; Lacroix et al., 2020; Bertin et al., 2023) in shallow, well-mixed water columns. On continental shelves, the outgassing of CO2 driven by the saturation of surface waters with terrestrial DIC or remineralized terrestrial organic carbon can also be compensated by the excess of alkalinity relative to DIC concentration (Cai, 2011; Louchard et al., 2021). In the absence of transformation in the coastal ocean, refractory riverine organic carbon can be transported offshore due to its slow turnover time (Hansell et al., 2004; Holmes et al., 2008; Kaiser et al., 2017). Concerning nutrients, their injection into the surface ocean can fertilize the growth of photosynthetic organisms in nutrient-limited regions. The subsequent primary production by photosynthetic organisms enhances CO2 uptake by carbon fixation. Globally, lateral inputs increase ocean primary productivity and contribute to an estimated coastal-ocean carbon sink of ~0.25 Pg C yr-1, which is roughly 10% to 35% of the global-ocean sink (Dai et al., 2022; Resplandy et al., 2024).

While monitoring global riverine inputs to the ocean is challenging due to the substantial financial/human effort, often in remote environments, land surface and watershed models can provide spatiotemporally-resolved lateral inputs at global scales (Mayorga et al., 2010; Krinner et al., 2005; Hagemann and Dümenil, 1997; Hagemann and Gates, 2003; Li et al., 2017; Bloom et al., 2020; Gao et al., 2023). Coupled with Global-Ocean Biogeochemical Models (GOBMs), it is thus possible to quantify the response of the coastal- and open-ocean carbon cycle to lateral inputs (Aumont et al., 2001; Lacroix et al., 2021; Mathis et al., 2022; Louchard et al., 2021; da Cunha and Buitenhuis, 2013; Le Fouest et al., 2013; Terhaar et al., 2019; Gao et al., 2023; Bertin et al., 2023; Manizza et al., 2019; Séférian et al., 2020). Here, we extend the ECCO-Darwin state estimate by implementing global point-source lateral inputs of carbon and nutrients. ECCO-Darwin combines (i) property-conserving circulation from the Estimating the Circulation and Climate of the Ocean (ECCO) project, (ii) the MIT Darwin Project's marine ecology model, (iii) ocean carbon chemistry, and (iv) data assimilation tools developed by ECCO. The system provides global, data-constrained estimates of circulation, sea ice, ecology, and biogeochemistry, with demonstrated skill in reproducing variability in the carbon cycle (Carroll et al., 2020; Carroll et al., 2022; Bertin et al., 2023). However, prior studies have focused mainly on pelagic processes and have not included lateral fluxes of carbon and nutrients.

In this study, we 1) add point-source lateral inputs of carbon and nutrients to ECCO-Darwin globally and 2) evaluate the model response of air-sea CO2 flux and primary production to riverine biogeochemical inputs during 2000–2019. Implementing biogeochemical river inputs into ECCO-Darwin is a key stepping stone in the development of this data-constrained modeling framework. The sensitivity analysis described herein will allow for further integration of the Land-Ocean Aquatic Continuum (LOAC) in a fully-optimized ECCO-Darwin solution; a pivotal pathway for understanding the response of ocean biogeochemistry to terrestrial inputs."

The methodologies section is missing the component of describing general nutrient an carbon sources and losses in the model, which I think is highly relevant to understanding the role of river inputs and their fate in the ocean. Where was biogeochemical matter originating from previous to the river implementation. How does the loss through burial work in the model? Maybe this is less relevant for ECCO-Darwin due to corrections made through assimilation, but it should still be described.

We added a paragraph in the Methods section, L79:

"In the absence of lateral fluxes, carbon in ECCO-Darwin is removed from the ocean through a combination of biological, chemical, physical, and air-sea exchange processes. Phytoplankton uptake of DIC during photosynthesis reduces upper-ocean carbon and forms organic matter, some of which sinks out of the mixed layer as export production. Additional CO2 drawdown occurs when surface waters are undersaturated relative to the atmosphere, leading to net air—sea CO2 uptake. Carbonate chemistry processes, such as precipitation and dissolution, modify alkalinity and buffer the partitioning of carbon species, thereby influencing surface-ocean DIC concentrations. Finally, physical transport through upwelling, mixing, subduction, and advection transports both DIC and organic carbon through the water column. Nutrients are supplied by upwelling and vertical mixing, consumed by phytoplankton growth, regenerated during remineralization, and exported with sinking organic matter collectively regulating the efficiency of carbon uptake and storage. In the water column, particulate matter (detritus, inorganic carbon, and living phytoplankton and zooplankton) sinks at prescribed velocities and is removed at the ocean bottom to limit the accumulation of particulates on the seafloor."

The river implementation accounts for present-day river fluxes as inputs, and the authors use these fluxes to spinup the model, if I understand correctly. These fluxes have however been subject to strongly dynamical changes over the past century. Due to the longtime scales in the ocean, most of the ocean is thus likely more closely equilibrated with preindustrial fluxes. Thus spinning up the model with present-day

fluxes will likely overestimate their contributions in the open ocean (which the authors explain due to offshore transport processes solely). I think this should definitely be discussed and taken account for future work.

The CO2 source simulated by the model is strongly underestimated compared to what is currently estimated and used in the Global Carbon Budget (Regnier et al., 2022; Friedlingstein et al., 2024). The magnitudes of C, N and alkalinity river inputs, seem however consistent with literature values, which are thought to favor CO2 outgassing taking account input-to-export/burial ratios. Could the authors potentially better explain the discrepencies of the flux simulated in the model with presently used values (Is it a question of timescale of the spinup? Assuming present-day atmospheric CO2 versus preindustrial concentrations? Stoichiometries?).

Regnier, P., Resplandy, L., Najjar, R.G. et al. The land-to-ocean loops of the global carbon cycle. Nature 603, 401–410 (2022). https://doi.org/10.1038/s41586-021-04339-9

As a data-assimilative model, ECCO-Darwin does not use a traditional spin-up. The model does require a short adjustment period (about 3 years) to equilibrate due to a mismatch between initial conditions and observational constraints. Adding riverine fluxes extended the period of equilibrium, so we used outputs from 2000 onward. Through the assimilation of observations from 1992–2019, the ocean biogeochemical state simulated by ECCO-Darwin is actually representative of past and present conditions and the anthropogenically-perturbed natural river loop. Consequently, the optimized version of ECCO-Darwin used in Baseline already implicitly accounts for lateral fluxes implicitly in its data-constrained state estimate. However, we were expecting a more significant response from the present sensitivity analysis.

We agree with Referee #1's comment and have adjusted and qualified the text accordingly. We discuss our estimate of air-sea CO2 flux changes in response to present-day riverine biogeochemical fluxes (in light of estimates of the river-loop) and we now clarify this in the revised manuscript that our model set-up is not relevant for representing the pre-industrial river-loop, L294:

"As our estimated global-ocean  $CO_2$  uptake in Baseline (-2.58 Pg C yr-1) explicitly accounts for air-sea  $CO_2$  flux from modern rising atmospheric  $CO_2$  concentration and climate change, the addition of riverine inputs should close the gap between the global-ocean  $CO_2$  uptake from Baseline and from present-day estimates of the global-ocean  $CO_2$  sink that consider the river-loop (-1.85±0.95 Pg C yr-1; ±2 $\sigma$ ) — the natural river loop being responsible for a  $CO_2$  outgassing of 0.65±0.3 Pg C yr-1 (±2 $\sigma$ ) (Regnier et al., 2022). In the simulation with riverine carbon only (DCrun), our estimate of +0.22 Pg C yr-1 of air—sea  $CO_2$  flux is lower than previous preindustrial-based estimates of +0.59 Pg C yr-1 (Aumont et al., 2001) and +0.65 Pg C yr-1 (Regnier et al., 2022), but within the same order of magnitude. When including both riverine carbon and nutrients (ALLrun), our model simulates a smaller increase in air—sea  $CO_2$  flux (+0.02 Pg C yr-1),

alongside a positive NPP response (+1 Pg C yr-1). This differs from the pre-industrial estimates of Lacroix et al. (2020), who found a comparable increase in air–sea CO2 flux (+0.23 Pg C yr-1) but a reduction in NPP (-1.78 Pg C yr-1) due to stabilizing ocean biogeochemical inventories. Adding nutrient inputs increases global-ocean marine NPP by 1 Pg C yr-1 compared to Baseline. The addition of  $t_{DIN}$  and  $t_{DON}$  also increased ocean NPP by 0.6 Pg C yr-1 in the model described by Tivig et al. (2021) (Table 4). In our study, the increase in NPP per surface area driven by riverine inputs was stronger in the coastal ocean compared to the open ocean, relative to their respective surface areas. This is consistent with the recent study of Mathis et al. (2024), which demonstrates the role of increased nutrient inputs in driving stronger biological carbon fixation and, thus, an enhanced CO2 sink in the coastal ocean during the last century. We note that our multi-decadal estimates do not reach equilibrium following the addition of riverine inputs (Figures S2–S9) and do not have a realistic representation of blue carbon, bottom-sediment processes, and fine-resolution coastal ecosystems that drive the coastal-ocean sink and transformation of elements. Therefore, our results are not directly comparable to long-term and pre-industrial estimates of the ocean response to riverine inputs (Regnier et al., 2022; Resplandy et al., 2024)."

I would suggest additionally using regional coastal water residence times estimated in literature and their differences between regions to explain differences for the different coastal responses in different regions of focus (e.g. regional values in Liu et al., 2019; Lacroix et al., 2021). For instance, high residence times in Arctic region could favor more outgassing of terrestrial organic material, versus less uptake in these light-limited regions through the nutrient inputs.

Liu, X., Dunne, J. P., Stock, C. A., Harrison, M. J., Adcroft, A., & Resplandy, L. (2019). Simulating Water Residence Time in the Coastal Ocean: A Global Perspective. Geophysical Research Letters, 46, 13910–13919. https://doi.org/10.1029/2019GL085097

Lacroix, F., Ilyina, T., Laruelle, G. G., & Regnier, P. (2021). Reconstructing the preindustrial coastal carbon cycle through a global ocean circulation model: was the global continental shelf already both autotrophic and a CO2 sink?. Global Biogeochemical Cycles, 35, e2020GB006603. <a href="https://doi.org/10.1029/2020GB006603">https://doi.org/10.1029/2020GB006603</a>

This is a great suggestion, and we now discuss our results in light of coastal water residence time. It allowed us to further highlight the strengths and limits of our approach. Our model results for the Arctic are consistent with this mechanism, as we find strong CO2 outgassing in the region when riverine carbon and nutrients are included. This is consistent with the relatively long residence times of Arctic shelf waters, which favor the remineralization and outgassing of terrestrial organic material, while nutrient-driven uptake is primarily limited by light. It is also consistent with Referee

**2's comment on equilibrium time after adding riverine inputs. The Arctic's long residence time prolongs adjustment to riverine perturbations, explaining why this region is slower to equilibrate compared to more rapidly flushed systems.**

In contrast, for regions with shorter residence times, one would expect less potential for remineralization and outgassing. However, our simulations still show substantial CO2 outgassing in some of these regions, such as in the Amazon plume, but are logically accompanied by increased offshore transport. This discrepancy may indicate that our method is overestimating the amount of river-derived carbon delivered and being processed in these coastal zones, particularly where rapid flushing reduces the residence time of riverine material. L273:

"The 28-year model period (1992–2019) does not allow the system to fully equilibrate with the addition of riverine inputs. However, time series of change in air–sea CO2 flux and NPP with the addition of river carbon and nutrients (Supporting Information Figures S2–S9) indicate that most regions approach quasi-equilibrium by the year 2000, consistent with the global response. In contrast, the change in air-sea CO2 flux and NPP with the addition of river carbon and nutrients in the Arctic do not stabilize over the model period (Supporting Information Figures S3 and S7). Regional variability in air-sea CO2 flux responses can be interpreted through differences in coastal residence times, as in the Arctic, long residence times promote remineralization and outgassing of terrestrial organic matter while limiting nutrient-driven uptake due to light limitation (Liu et al., 2019; Lacroix et al., 2021a). These extended residence times also explain why the Arctic response does not stabilize within the 28-year experiment timescale (Supporting Information Figures S3 and S7), in contrast to other regions where shorter residence times facilitate more rapid equilibration. Conversely, regions such as the Amazon plume display substantial CO2 outgassing despite shorter residence times, but this is accompanied by elevated offshore transport, suggesting that riverine carbon inputs or remineralization rates may be overestimated in coastal systems where residence time is short."

**Specific Comments**

L17 "At the same time, most of the refractory part of riverine organic carbon is transported offshore from river mouth regions as it is remineralized at slower turnover rates." This is a bit of a jump from the previous sentence. Did you mean to add that river transports play a central role for biogeochemical processes in the coastal ocean first?

We changed this sentence to make it clearer and moved it to the Introduction second paragraph about terrestrial carbon and nutrient fate, L23:

"In the absence of transformation in the coastal ocean, refractory riverine organic carbon can be transported offshore due to its slow turnover time (Hansell et al., 2004; Holmes et al., 2008; Kaiser et al., 2017)."

L25 I would also add that terrestrial OC is thought to cause a source of CO2 to the atmosphere (after degradation).

Done. We added this to the sentence on L21:

"On continental shelves, the outgassing of CO2 driven by terrestrial DIC or remineralized terrestrial organic carbon can also be compensated by the excess of alkalinity relative to DIC concentration (Cai, 2011; Louchard et al., 2021)."

L27 "Globally, this lateral input increases ocean primary productivity and contributes to an estimated coastal-ocean sink of~ 0.25 Pg C yr-1, which is roughly 17% of the global-ocean sink (Cai, 2011; Lacroix et al., 2021; Gao et al., 2023)." I would use Dai et al. (2022) and Resplandy et al. (2024) here as more recent estimates:

Dai, M., Su, J., Zhao, Y., Hofmann, E. E., Cao, Z., Cai, W. J., ... & Wang, Z. (2022). Carbon Fluxes in the Coastal Ocean: Synthesis, Boundary Processes, and Future Trends. *Annual Review of Earth and Planetary Sciences*, *50*, 593-626.

Resplandy, L., Hogikyan, A., Müller, J. D., Najjar, R. G., Bange, H. W., Bianchi, D., ... & Regnier, P. (2024). A Synthesis of Global Coastal Ocean Greenhouse Gas Fluxes. *Global* Biogeochemical Cycles, 38(1), e2023GB007803.

Done. We added these references and adjusted the range to account for the new estimates from Resplandy et al., L27:

"Globally, lateral inputs increase ocean primary productivity and contribute to an estimated coastal-ocean sink ranging from 0.2–0.7 Pg C yr-1, which is roughly 10% to 35% of the global-ocean sink (Dai et al., 2022; Resplandy et al., 2024)."

L157 "Additionally, the data-based products exhibited lower surface-ocean pCO2 compared to ECCO-Darwin Baseline (Figure 1i) in the Arctic Ocean and near the periphery of Antarctica; regions where observations are highly limited in space and time." Could this potentially also be a sea ice representation problem?

Data-products use the ice-free fraction to compute  $pCO_2$  and air-sea  $CO_2$  fluxes. We discarded grid cells covered by sea ice (concentration > 0%) from the model-data evaluation, based on the percentage of sea-ice cover simulated by ECCO-Darwin. A known bias in ECCO-Darwin compared to data-based products is the tendency to simulate stronger winter  $CO_2$  uptake in subpolar regions due to the interplay of surface

cooling and mixed layer DIC divergence (Lauderdale et al., 2016; Carroll et al., 2020), which could reflect limitations in the model's representation of seasonal mixed layer dynamics. Similar issues, including uncertainties in sea-ice representation skill, may contribute to the discrepancies we report in the Arctic and Southern Ocean, where observational coverage is sparse. We added this to the discussion L238:

"This mismatch in the Arctic and in Antarctica may also reflect known biases in ECCO-Darwin's representation of the seasonal mixed layer and sea-ice dynamics (Lauderdale et al., 2016; Carroll et al., 2020)."

L193 "the increase of NPP" -> The areal increase in NPP?

Yes, fixed.

L197 "In Baseline, ARCT results in a CO2 uptake of roughly 0.21 Pg C yr-1." This reads as if ARCT was a simulation, would slightly revise the wording.

Done, changed it to

"In Baseline, the CO2 uptake in ARCT was roughly 0.21 Pg C yr-1."

Table 4. In terms of global FCO2, I would add the estimates of Aumont et al., 2002 and Lacroix et al., 2020. The table also only shows model derived estimates, whereas some budget-derived estimates also exist and are, as of now, preferably used in assessments (e.g. Regnier et al., 2022).

Aumont, O., J. C. Orr, P. Monfray, W. Ludwig, P. Amiotte-Suchet, and J.-L. Probst (2001), Riverine-driven interhemispheric transport of carbon, Global Biogeochem. Cycles, 15(2), 393–405, doi:10.1029/1999GB001238.

We voluntarily kept these numbers out of Table 4 as they correspond to pre-industrial estimates and/or omit the presence of terrestrial nutrients. However, we added a paragraph that discusses these global estimates L295:

"In this study, carbon inputs drive a  $CO_2$  outgassing of 0.22 Pg C yr-1, while nutrient inputs drive a  $CO_2$  uptake of 0.20 Pg C yr-1 from enhanced primary productivity, which primarily occurs in the coastal ocean. Combined, carbon and nutrients inputs in  $ALL_{run}$  are limited to a 0.02 Pg C yr-1  $CO_2$  outgassing; lower than estimates from the literature (Table 4). As our estimated global-ocean  $CO_2$  uptake in Baseline (-2.58 Pg C yr-1) explicitly accounts for air-sea  $CO_2$  flux from modern rising atmospheric  $CO_2$  concentration and climate change, the addition of riverine inputs should close the gap between the global-ocean  $CO_2$  uptake from Baseline and from present-day estimates of the global-ocean  $CO_2$  sink that consider the river-loop (-1.85±0.95 Pg C yr-1; ±2 $\sigma$ ) — the

natural river loop being responsible for a CO2 outgassing of 0.65±0.3 Pg C yr-1 (±2σ) (Regnier et al., 2022). In the simulation with riverine carbon only (DCnin), our estimate of +0.22 Pg C yr-1 of air–sea CO2 flux is lower than previous preindustrial-based estimates of +0.59 Pg C yr-1 (Aumont et al., 2001) and +0.65 Pg C yr-1 (Regnier et al., 2022), but within the same order of magnitude. When including both riverine carbon and nutrients (ALLrun), our model simulates a smaller increase in air–sea CO2 flux (+0.02 Pg C yr-1), alongside a positive NPP response (+1 Pg C yr-1). This differs from the pre-industrial estimates of Lacroix et al. (2020), who found a comparable increase in air-sea CO2 flux (+0.23 Pg C yr-1) but a reduction in NPP (-1.78 Pg C yr-1) due to stabilizing ocean biogeochemical inventories. Adding nutrient inputs increases global-ocean marine NPP by 1 Pg C  $yr^{-1}$  compared to Baseline. The addition of  $t_{DIN}$  and  $t_{DON}$  also increased ocean NPP by 0.6 Pg C yr-1 in the model described by Tivig et al. (2021) (Table 4). In our study, the increase in NPP per surface area driven by riverine inputs was stronger in the coastal ocean compared to the open ocean, relative to their respective surface areas. This is consistent with the recent study of Mathis et al. (2024), which demonstrates the role of increased nutrient inputs in driving stronger biological carbon fixation and, thus, an enhanced CO2 sink in the coastal ocean during the last century. We note that our multi-decadal estimates do not reach equilibrium following the addition of riverine inputs (Figures S2–S9) and do not have a realistic representation of blue carbon, bottom-sediment processes, and fine-resolution coastal ecosystems that drive the coastal-ocean sink and transformation of elements. Therefore, our results are not directly comparable to long-term and pre-industrial estimates of the ocean response to riverine inputs (Regnier et al., 2022; Resplandy et al., 2024)."

---

## Author Comment (AC2)

We thank the referees for their constructive feedback. We agree with all major comments. Our study needed a clearer message and scope, along with a better assessment of its limitations, particularly regarding missing processes, spin-up, and stoichiometry. We now explicitly position this work as a sensitivity analysis using an existing optimized ECCO-Darwin to quantify the response of the ocean carbon cycle to riverine inputs, rather than to deliver a fully optimized solution that includes coastal processes.

To address concerns about nutrient stoichiometry and phytoplankton limitation, we streamlined the experiment set by adding a single dedicated nutrient simulation that includes both phosphorus and iron inputs. This configuration directly responds to referee feedback by accounting for their combined supply relative to global N:P:Si:Fe stoichiometry. Regarding model spin-up and equilibration timescales, we acknowledge that multi-decadal simulations cannot fully equilibrate riverine nutrient perturbations, especially in the Arctic, and we clarify that such equilibration is required for comparison with the long-term river loop. We further highlight that spin-up limitations will be directly addressed in future work, as we are currently developing a new adjoint-optimized ECCO-Darwin configuration (v06) that includes fully-coupled riverine biogeochemical tracers and other coastal and benthic processes.

In summary, we accept the referee's broader point that a more complete and optimized configuration of ECCO-Darwin will be necessary for comparison with long-term cycling of riverine inputs; this manuscript is an important technical milestone towards that effort Finally, while we recognize suggestions to further expand the analysis, we have deliberately chosen to keep the manuscript focused solely on the implementation of riverine inputs and avoid adding additional layers of complexity, as it is already dense. Our intention in submitting to GMD is for this work to serve as a technical foundation that can be leveraged for more comprehensive scientific exploration in future studies using ECCO-Darwin.

**Referee #2**

The manuscript describes impacts of adding observationally constrained rates of riverine input of DIC, DOC, DIN, DON and DSi to the global multi-functional type biogeochemical ECCO-Darwin model. The impact of various combinations of riverine carbon and nutrient supplies on model-data misfits with respect to surface pCO2 and air-sea CO2 fluxes are investigated, and some improvements are found, particularly when riverine DOC supply is simulated.

The study adds some information to other recent efforts to address the historical lack of realistic descriptions of the land-ocean interface in current global marine biogeochemical models. Results are relatively unsurprising, i.e. CO2 emissions from the ocean increase when DOC with a lifetime of 100 days is added to the ocean surface at river mouths, and CO2 uptake by the ocean increases in some regions where additional nutrients are added by rivers. The design of the study results in only limited gain in terms of scientific understanding. I have three main concerns about the present version of the study that limit the gain in scientific understanding:

First, the riverine input of biogeochemical tracers is added to a calibrated model run without riverine input, which -presumably- tries to make up for the missing river input by adjusting model parameters or other control variables. The model configuration with riverine input is not calibrated. Thus, the comparison is between a calibrated and an uncalibrated model version. Difficult to assess. Results are more a sensitivity analysis rather than an assessment of structural model improvements.

We agree that the present experiments should be interpreted as a sensitivity analysis rather than as a full optimization of ECCO-Darwin with riverine inputs. As a data-assimilative model, ECCO-Darwin adjusts biogeochemical control parameters and initial conditions to minimize model—data error, which implicitly accounts for some missing processes such as lateral fluxes. Through the assimilation of observations from 1992–2019, the ocean biogeochemical state simulated by ECCO-Darwin is actually representative of past and present conditions and the anthropogenically-perturbed natural river loop. Consequently, the optimized version of ECCO-Darwin used in Baseline already implicitly accounts for lateral fluxes. However, the coverage of datasets assimilated into ECCO-Darwin remains limited near the coast and river mouths, where terrestrial carbon and nutrients are mostly used or transformed, so a double accounting might be limited. Future efforts will focus on optimizing the model with river inputs, which we anticipate will further improve the structural realism of ECCO-Darwin and better constrain coastal processes. To better showcase the paper, we changed the title to:

Implementing Riverine Biogeochemical Inputs in ECCO-Darwin: A Sensitivity Analysis of Terrestrial Fluxes in a Data-Assimilative Global Ocean Biogeochemistry Model

Second, the simulations with riverine inputs are, if I understand correctly, run only from year 1992 to 2019, i.e. 28 years, of which the first 8 years are taken as spin-up, and 2000-2019 as analysis period. Particularly for the addition of nutrients, this is likely insufficient to reach a steady state. Some time series of relevant model output (NPP in different regions, nutrient and carbon concentrations) needs to be shown to allow the

reader to assess the issue of inferring general results from short decadal-scale simulations only.

We acknowledge that the 30-year model period (1992–2019) does not allow the system to fully equilibrate with the addition of river inputs, particularly the long-term river loop and nutrient inputs. We added Figures S3 and S4 so the reader can see that changes in air-sea CO2 flux and NPP in all experiments tend to stabilize from 2000, globally. We also computed these figures for our three regional domains in Figures S2–S9. While air-sea CO2 flux and NPP in TROP-ATL and SE-ASIA equilibrate around the same year as the global domain, they did not stabilize in ARCT, confirming Referee #2's suggestion. It is also consistent with Referee #1 comment on residence time. The Arctic's long residence time prolongs adjustment to riverine perturbations, explaining why this region is slower to equilibrate compared to regions with faster residence time. Extending the analysis over centennial timescales is an important next step, which we highlight in the revised discussion, L276:

"The 28-year model period (1992–2019) does not allow the system to fully equilibrate with the addition of riverine inputs. However, time series of change in air—sea CO2 flux and NPP with the addition of river carbon and nutrients (Supporting Information Figures S2-S9) indicate that most regions approach quasi-equilibrium by the year 2000, consistent with the global response. In contrast, the change in air-sea CO2 flux and NPP with the addition of river carbon and nutrients in the Arctic do not stabilize over the model period (Figures S3 and S7). Regional variability in air–sea CO2 flux responses can be interpreted through differences in coastal residence times, as in the Arctic, long residence times promote remineralization and outgassing of terrestrial organic matter while limiting nutrient-driven uptake due to light limitation (Liu et al., 2019; Lacroix et al., 2021a). These extended residence times also explain why the Arctic response does not stabilize within the 28-year experiment window (Figure S3 and S7), in contrast to other regions where shorter residence times facilitate more rapid equilibration. Conversely, regions such as the Amazon plume display substantial CO2 outgassing despite shorter residence times, but this is accompanied by offshore transport, suggesting that riverine carbon inputs or remineralization rates may be overestimated in coastal systems where residence time is short."

Third, several implicit and explicit assumptions may have relevant impacts on the results shown. Some assumptions are stated, e.g. the assumed 100 day lifetime of riverine DOC, or the assumption of zero phosphorus input from land while dissolve silica is included, others are not, such as the presence of denitrification and/or nitrogen fixation. In order to provide "a critical step forward", as stated in the title of the manuscript, a more comprehensive sensitivity analysis with respect to major

assumptions would be required, possibly extending the analysis to riverine supply of phosphorus.

We agree that our results rely on several key assumptions. In the Discussion section, we elaborate on the necessary and consequential simplifications made in this study. We also describe ongoing and future developments of ECCO-Darwin that will address these limitations and move toward a fully optimized ECCO-Darwin that accounts for coastal margins. To address concerns about nutrient stoichiometry and phytoplankton limitation, we streamlined the experiment set by adding a single dedicated nutrient simulation that includes nitrogen, phosphorus, silica, and iron inputs together. This configuration directly responds to referee feedback by accounting for their combined supply relative to global N:P:Si:Fe stoichiometry. We also agree that the global effect of the implementation is limited in this study. However, we consider the implementation critical for the ECCO-Darwin model and future land-to-ocean model development and studies. Until now, ECCO-Darwin lacked a realistic representation of lateral fluxes of carbon and nutrients, and including these sources is an important step forward for the ECCO-Darwin community, but also for the ocean modeling community in general, as more ocean biogeochemistry models should account for terrestrial carbon and nutrients. Even if the contribution of river carbon and nutrients is small at the global scale, it remains significant for coastal and regional budgets and, thus, is pivotal for future studies using ECCO-Darwin. For this reason, we believe it is critical to move forward with this new ECCO-Darwin capability. We changed the title to make it more reflective of both referees' comments:

Implementing Riverine Biogeochemical Inputs in ECCO-Darwin: A Sensitivity Analysis of Terrestrial Fluxes in a Data-Assimilative Global Ocean Biogeochemistry Model

**Individual points:**

**I.17 'slower' than what?**

To clarify, we changed this sentence in the new version of the introduction, L23:

"In the absence of transformation in the coastal ocean, refractory riverine organic carbon can be transported offshore due to its slow turnover time (Hansell et al., 2004; Holmes et al., 2008; Kaiser et al., 2017)."

I.24/25 'excess of alkalinity relative to DIC' Does this refer to concentrations or to fluxes? How does this fit to outgassing (a flux)?

To clarify, we changed this sentence in the new version of the introduction, L21:

"On continental shelves, the outgassing of CO2 driven by the saturation of surface waters with terrestrial DIC or remineralized terrestrial organic carbon can also be compensated by the excess of alkalinity relative to DIC concentration (Cai, 2011; Louchard et al., 2021)."

I.27 There does not always have to be alkalinity production, e.g. when calcifiers are involved.

We removed the mention of alkalinity production.

I.28 'estimated coastal-ocean sink' of what ? Total carbon, riverine carbon, marine carbon?

We changed it to: "coastal-ocean carbon sink"

I.84 does 'particulate organic matter' mean detritus or phytoplankton and zooplankton as well?

It includes detritus, inorganic carbon and living phytoplankton and zooplankton. We added this to the description, L88:

"In the water column, particulate matter (detritus, inorganic carbon, and living phytoplankton and zooplankton) sinks at prescribed velocities and is removed at the ocean bottom to limit the accumulation of particulates on the seafloor."

I.93. Does this mean there is only 8 years of spin-up? Is the biogeochemistry in some form of steady state after such a short period, and if so, in what regions?

As a data-assimilative model, ECCO-Darwin does not use a traditional spin-up. The model does require a short adjustment period (about 3 years) to equilibrate due to a mismatch between initial conditions and observational constraints. Adding riverine fluxes extended the period of equilibrium, so we used outputs from 2000 onward. We acknowledge that the 30-year model period (1992–2019) does not allow the system to fully equilibrate with the addition of river inputs, particularly the long-term river loop and nutrient inputs. We added Figures S3 and S4 so the reader can see that changes in air-sea CO2 flux and NPP in all experiments tend to stabilize from 2000, globally. We also computed these figures for our three regional domains in Figures S5-S11. While air-sea CO2 flux and NPP in TROP-ATL and SE-ASIA equilibrate around the same year as the global domain, they did not stabilize in ARCT, confirming Referee #2's suggestion in their second major comment. We added the importance of equilibration in the discussion, L273:

"The 28-year model period (1992–2019) does not allow the system to fully equilibrate with the addition of riverine inputs. However, time series of change in air—sea CO2 flux and NPP with the addition of river carbon and nutrients (Supporting Information Figures S3–11) indicate that most regions approach quasi-equilibrium by the year 2000, consistent with the global response. In contrast, the change in air-sea CO2 flux and NPP with the addition of river carbon and nutrients in the Arctic do not stabilize over the model period (Supporting Information Figures S3-S7). Regional variability in air-sea CO2 flux responses can be interpreted through differences in coastal residence times, as in the Arctic, long residence times promote remineralization and outgassing of terrestrial organic matter while limiting nutrient-driven uptake due to light limitation (Liu et al., 2019; Lacroix et al., 2021a). These extended residence times also explain why the Arctic response does not stabilize within the 28-year experiment window (Supporting Information Figure S3 and S7), in contrast to other regions where shorter residence times facilitate more rapid equilibration. Conversely, regions such as the Amazon plume display substantial CO2 outgassing despite shorter residence times, but this is accompanied by offshore transport, suggesting that riverine carbon inputs or remineralization rates may be overestimated in coastal systems where residence time is short."

I.96ff The model evaluation addresses surface pCO2 and air-sea fluxes of CO2 only. It would be useful to provide some assessment of simulated NPP, biomass and nutrient distributions.

As ECCO-Darwin is a data-assimilative model, it matches very well the nutrient distributions compared to observational datasets used for data constraint (see Figure 2 in Carroll et al., 2020). Our NPP estimate remains in the lower range of literature estimates (30–75 Pg C yr-1). As NPP estimates in literature come with large uncertainties, we preferred to discuss and compare our simulated NPP with existing estimates in text only L254. However, we added a figure that compares ECCO-Darwin Baseline with two state-of-the-art references: the ICON-Coast ocean model (Mathis et al., 2022) and the MODIS CAFE remote sensing algorithm (Silsbe et al., 2016).

"Baseline captures similar spatial patterns of NPP compared to the model ensemble of the REgional Carbon Cycle Assessment and Processes Phase-2 (RECCAP-2) project that aims at constraining present-day ocean carbon from observation-based estimates, inverse models, and GOBMs (Doney et al., 2024) (Supporting Information Figure S12). Many uncertainties remain regarding global-ocean NPP estimates from remote sensing (due to uncertainty in algorithms) and models (due to different conceptual model architectures). Overall, NPP in Baseline (24.5 Pg C yr-1) lies in the lower bound of the wide range depicted by the RECCAP-2 model ensemble (25–57 Pg C yr-1; Doney et al.,

2024) and remote-sensing algorithms (43–68 Pg C yr-1; Behrenfeld and Falkowski, 1997; Silsbe et al., 2016; Carr et al., 2006; Marra et al., 2003; Behrenfeld et al., 2005). This relatively low NPP results primarily from strong iron limitation in the High-Nutrient, Low-Chlorophyll (HNLC) regions in ECCO-Darwin (Carroll et al., 2020). The strong surface-ocean stratification and the weaker winter convection limit the replenishment of nutrients in the euphotic zone. Nevertheless, global-ocean NPP estimates will improve from enhanced space-time coverage of NPP measurements and associated key variables such as chlorophyll, light, nutrients, optical properties, and cell physiology (Bendtsen et al., 2023). An integration of environmental variables along with NPP measurements will greatly reduce models' spread and mismatch with synoptic in-situ observations. The implementation of a radiative transfer package (Dutkiewicz et al., 2019) in the next version of ECCO-Darwin, for which development is already underway, will permit the assimilation of direct ocean-color observations (remotely-sensed reflectance) and improve the model's estimate of global-ocean NPP."

Carroll, D., Menemenlis, D., Adkins, J. F., Bowman, K. W., Brix, H., & Dutkiewicz, S., et al. (2020). The ECCO-Darwin data-assimilative global ocean biogeochemistry model: Estimates of seasonal to multidecadal surface ocean pCO2 and air-sea CO2 flux. Journal of Advances in Modeling Earth Systems, 12, e2019MS001888. https://doi.org/10.1029/2019MS001888

I.138 Would be good to add if the extreme value was high or low, and also provide a very brief explanation for why (only) this value had to be corrected.

The value was originally about 9 Tmol C yr-1 of DIC coming from the Amazon River. It is due to the outstandingly large Amazon watershed area (used for estimating rock weathering) and freshwater discharge compared to other basins that drive a very high load when using equation 9 from Li et al. (2017). We added more details in the manuscript, L150:

"Due to overestimated  $t_{DIC}$  inputs in our Global NEWS 2-derived computation for the Amazon River,  $t_{DIC}$  inputs for this system were set to a more realistic, literature-mean of 2.54 Tmol C yr-1 (da Cunha and Buitenhuis, 2013; Probst et al., 1994; Li et al., 2017) (for more details, see Appendix A). The outstandingly large Amazon watershed area (used for estimating rock weathering) and freshwater discharge compared to other basins drive a very high DIC load when using equation 9 from (Li et al., 2017)".

Table 2: The units are unclear and likely wrong. If Tg/yr, then the assumed stoichiometry of at least DOC, DIN needs to be provided. References provided in Table 2 are too generic, e.g. some labeled 1-3 do not even mention DSi.

We changed the units to Tg C/yr, Tg N/yr, Tg P/yr and Tg Si/yr so it is less confusing for the reader and moved the references into separate rows. We also removed confusing references and any references that compute inputs for the pre-industrial era e.g. Lacroix et al., (2020) or that use Global NEWS 2 as a source for their estimates.

Fig.3: color scale does not seem optimal. A log-scale might allow easier interpretation.

Given that the colorbar contains negative and positive values, we did not use a log scale. Instead, we narrowed the bounds of the colorbar to make the changes more apparent.

**Table 3. Air-sea CO2 fluxes seem to have the wrong sign**

We used the common nomenclature for air-sea CO2 fluxes with negative sign describing an uptake by the ocean and positive sign for outgassing to the atmosphere (DeVries et al., 2023; Fay et al., 2024; Gregor et al., 2024). To avoid confusion, the direction of the fluxes is now mentioned in the caption of the table.

DeVries, T., Yamamoto, K., Wanninkhof, R., Gruber, N., Hauck, J., Müller, J. D., et al. (2023). Magnitude, trends, and variability of the global ocean carbon sink from 1985 to 2018. Global Biogeochemical Cycles, 37, e2023GB007780. <a href="https://doi.org/10.1029/2023GB007780">https://doi.org/10.1029/2023GB007780</a>

Fay, A. R., Carroll, D., McKinley, G. A., Menemenlis, D., & Zhang, H. (2024). Scale-dependent drivers of air-sea CO2 flux variability. Geophysical Research Letters, 51, e2024GL111911. https://doi.org/10.1029/2024GL111911

Gregor, L., Shutler, J., & Gruber, N. (2024). High-resolution variability of the ocean carbon sink. Global Biogeochemical Cycles, 38, e2024GB008127. https://doi.org/10.1029/2024GB008127

I.186. Presumably t\_DIC plays only a small role compared to t\_DOC because of the assumed compensation to DIC input by ALK input? Might be good to say this here.

This sentence was removed of the manuscript following the simplification of the experiments.

I.224 'and freshwater discharge' is misleading. If I understand correctly, freshwater discharge is identical in all simulations?

The sentence is indeed confusing and we changed it to, L221:

"The largest differences occur along the coastal periphery, especially near large river mouths, where Baseline's lack of riverine carbon and nutrients underestimates surface-ocean pCO2 compared to the data-based products."

1.235 'of dissolved carbon input' - should it read dissolved organic carbon input?

In this sentence, we compared data-based products with  $ALL_{run}$  and this pattern simulated near large river plumes is driven by  $DC_{run}$ , which includes both dissolved organic and inorganic carbon. We changed the sentence to clarify this, L232:

"In ALLrun, the deviation of ECCO-Darwin surface ocean pCO2 and air-sea CO2 fluxes from the data-based products is reduced near large river plumes (Amazon, Paraná, Congo, Ganges, Yangtze, Amur) by the addition of dissolved carbon inputs."

1.237 please provide a brief explanation of why model skill decreases here.

Done. We provide an explanation for each region in the following sentences, L234:

"However, in the Bay of Bengal, Sea of Japan, Canadian Arctic Archipelago, Caribbean Sea, and Siberian Shelf, model skill decreases with riverine inputs. While the data-based products converge on a CO2 sink over the Arctic basin, adding dissolved carbon in ALLrun and the associated CO2 outgassing increases the model mismatch in this region. However, large gaps in SOCAT data coverage in the Arctic Ocean may be responsible for this discrepancy, especially on the Siberian Shelf. This mismatch in the Arctic and in Antarctica may also reflect known biases in ECCO-Darwin's representation of the seasonal mixed layer and sea-ice dynamics (Lauderdale et al., 2016; Carroll et al., 2020). In the Bay of Bengal and Sea of Japan, the ocean CO2 sink is increased by the higher uptake from enhanced-NPP in response to the addition of nutrient inputs in ALLrun. In the Caribbean Sea, the slight increase in mismatch is associated with carbon inputs from Tropical Atlantic rivers, such as the Amazon and Orinoco."

I.248 why do you think input might be overestimated rather than underestimated? Could there be positive feedbacks, for example via redox-sensitive Fe and P cycling?

This is a great point, and LOAC can act in both directions by increasing or decreasing the amount of bioavailable material. Thus, we changed the sentence to, L253:

"In these regions, biogeochemical inputs might be overlooked due to the absence of LOAC processes in the model (i.e., parameterizations of estuarine mixing and biogeochemical processes) and the lack of a more realistic representation of organic matter remineralization, allowing for the advection of excess dissolved carbon and nutrients into the open ocean."

I.282ff would be good to mention possible effects of explicitly accounting for denitrification, as done in some previous studies cited by the authors.

Done. We added a mention to denitrification in estuaries, L330:

"The absence of denitrification within estuaries (3–10 Tg N yr-1; Seitzinger et al., 2005) could alter N:P stoichiometry and downstream air—sea CO2 fluxes. However, our results do not include air-sea CO2 fluxes associated with these land-to-ocean components."

Seitzinger, S. P., Mayorga, E., Bouwman, A. F., Kroeze, C., Beusen, A. H. W., Billen, G., Drecht, G. V., Dumont, E., Fekete, B. M., Garnier, J., and Harrison, J. A.: Global river nutrient export: A scenario analysis of past and future trends, Global Biogeochem. Cy., 24, GB0A08, https://doi.org/10.1029/2009GB003587, 2010.

**I.296 why overestimated and not underestimated?**

The referee is correct. We referred to the excess of advected riverine inputs but the response of the ocean carbon cycle can be either underestimated or overestimated. To clarify, we changed the sentence to, L314:

"As ALLrun deviates more from the data-based products in terms of surface-ocean pCO2 and air-sea CO2 flux in TROP-ATL, ARCT, and SE-ASIA compared to Baseline, the response of the ocean carbon cycle (source and uptake) to riverine inputs might be inaccurate due to the advection of excess elements to the open ocean."

**I.297 why only faster degradation and not slower?**

Here, we mention only faster degradation as it refers to an excess of material that reaches the open ocean. With faster degradation, this excess would be reduced.

I.332 I do not understand 'lack of nitrogen and silica-limited taxa' The model explicitly resolves diatoms and 4 other phytoplankton species. Aren't diatoms nitrogen and silica-limited in your model?

We agree that this is confusing. We referred to diatom species representative of the Arctic as ECCO-Darwin global configuration uses diatoms' traits that are not region specific (i.e., a global ecosystem). To clarify, we changed the sentence to, L351:

"We stress that the phytoplankton functional types in our global model are not representative of the specific Arctic Ocean ecology, and the lack of regionally-adjusted affinity for specific nutrients might hinder the model ecosystem response to riverine nutrients (Ardyna and Arrigo, 2020)."

I.362. I do see that this 'study is a critical step forward'. This would have to be justified in more detail.

We agree that we need to emphasize more the critical aspect of this study. This is addressed in the general response to both referees.

**I.403 why only overestimated and not underestimated?**

Agree. This goes in both directions. To clarify, we changed the sentence on L410 to:

"Overall, this could lead to an inexact  $t_{DOC}$  remineralization in some regions and thus excess of either ocean  $CO_2$  outgassing due to an excess of DIC or advection of organic matter to the open ocean; a limitation that also exists in other GOBMs due to undifferentiated remineralization rates."